# END-TO-END TOPOGRAPHIC AUDITORY MODELS REPLICATE SIGNATURES OF HUMAN AUDITORY CORTEX

## ABSTRACT

The human auditory cortex is topographically organized. Neurons with similar response properties are spatially clustered, forming smooth maps for acoustic features such as frequency in early auditory areas, and modular regions selective for music and speech in higher-order cortex. Yet, evaluations for current computational models of auditory perception do not measure whether such topographic structure is present in a candidate model. Here, we show that cortical topography is not present in the previous best-performing models at predicting human auditory fMRI responses. To encourage the emergence of topographic organization, we adapt a cortical wiring-constraint loss originally designed for visual perception. The new class of topographic auditory models, TopoAudio, are trained to classify speech, and environmental sounds from cochleagram inputs, with an added constraint that nearby units on a 2D cortical sheet develop similar tuning. Despite these additional constraints, TopoAudio achieves high accuracy on benchmark tasks comparable to the unconstrained non-topographic baseline models. Further, TopoAudio predicts the fMRI responses in the brain as well as standard models, but unlike standard models, TopoAudio develops smooth, topographic maps for tonotopy and amplitude modulation (common properties of early auditory representation, as well as clustered response modules for music and speech (higher-order selectivity observed in the human auditory cortex). TopoAudio is the first end-to-end biologically grounded auditory model to exhibit emergent topography, and our results emphasize that a wiring-length constraint can serve as a general-purpose regularization tool to achieve biologically aligned representations.

## 1    INTRODUCTION

The human auditory cortex has a well documented *topographic* organization in which neurons with similar response properties are spatially clustered (Moerel et al., 2014; Brewer & Barton, 2016; Scheich, 1991; Kanold et al., 2014; Read et al., 2002; Leaver & Rauschecker, 2016). In early auditory areas, this organization gives rise to smooth topographic maps for acoustic features such as frequency (tonotopy), amplitude modulation, and pitch (Reale & Imig, 1980; Bendor & Wang, 2005; Wessinger et al., 1997; Allen et al., 2022; Joris et al., 2004; Baumann et al., 2015; Norman-Haignere et al., 2013). In higher-order regions, distinct clusters emerge for more complex categories like music and speech (Zatorre et al., 2002; Leaver & Rauschecker, 2010; Norman-Haignere et al., 2015; 2022; Harris et al., 2023; Williams et al., 2022; Fedorenko et al., 2012; Boebinger et al., 2021). In recent years, computational models have emerged that capture key aspects of human auditory behavior (Francl & McDermott, 2022; Saddler et al., 2020; 2021; Saddler & McDermott, 2024; Koumura et al., 2023) and predict neural responses (Kell et al., 2018; Tuckute et al., 2023; Giordano et al., 2023; Güçlü et al., 2016; Khatami & Escabí, 2020; Vaidya et al., 2022; Millet et al., 2022; Rupp et al., 2025). Yet even the leading models of the auditory cortex (Tuckute et al., 2023) neither incorporate, evaluate, nor explain the topography observed in the brain.

To develop such models, we take inspiration from recent work that utilizes wiring-length constraints to induce topographic structure. One compelling hypothesis for the ubiquitous nature of cortical maps is that they emerge from optimizing task representations under wiring-length constraints (Kaas, 1997; Chklovskii & Koulakov, 2004; Jacobs & Jordan, 1992). If wiring-length indeed serves

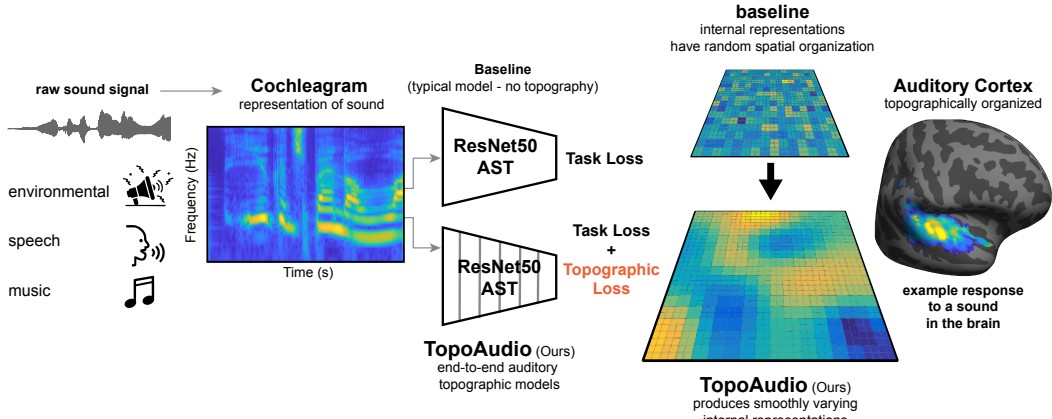

Figure 1: **TopoAudio: Topographic audio models.** Raw sound waveforms are transformed into cochleagrams, which serve as inputs to the neural network backbone (middle). The baseline model (middle, top) is trained using only a task loss and learns spatially disorganized internal representations. In contrast, our proposed model TopoAudio is trained with an additional spatial smoothness constraint via TopoLoss. This results in smooth, topographically organized representations that resemble those observed in human auditory cortex (right).

as a domain-general principle in neural representation, then it should suffice to capture cortical maps in any modality. This principle was recently implemented in vision models Lee et al. (2020); Margalit et al. (2024); Qian et al. (2024); Deb et al. (2025), where a wiring-length constraint led to an emergence of topographic information and a better alignment with neural data. More recently, in attempt to show generality, a wiring-length constraint was applied to language models (Rathi et al., 2025; Deb et al., 2025), however the overall topography of core language regions in the biological brain is debated, so it is unclear if these results lead to a more biological model. The auditory domain is a natural test of this wiring-length hypothesis – it encodes dynamic features, and has some known topographic structure.

In this work, we present **TopoAudio**, the first suite of end-to-end topographic auditory models. Our models achieve competitive performance at predicting fMRI responses in the human auditory cortex and perform at par with baseline models on auditory classification tasks. Beyond these standard evaluations, we further investigate if the wiring-length constraint leads to hallmark features of auditory cortical organization: smooth tonotopic and modulation maps, and spatially clustered regions selective for speech and music. We find that the internal representations of TopoAudio models are more geometrically aligned with human brain data than standard models, demonstrating that incorporating topographic structure yields models that are both high-performing and biologically grounded. Our TopoAudio models provide strong evidence of a domain-general principle for introducing topography into AI models, enabling them to capture the emergence of spatial organization in any modality.

## 2 RELATED WORKS

### 2.1 TOPOGRAPHIC MODELS IN VISION AND LANGUAGE.

Topography is ubiquitous across sensory cortices. Perhaps the best known example is the visual cortex, where in early visual cortex neurons are arranged in fine-scale maps for orientation preference (Blasdel, 1992) - forming structured motifs such as pinwheels. At larger scales, cortical maps exhibit biases for real-world properties like object size, animacy (Konkle & Oliva, 2011; Konkle & Caramazza, 2013), and eccentricity, including specialized regions selective for faces, bodies, scenes, and words (Kanwisher, 2000; Grill-Spector et al., 2004; Epstein et al., 1999; Downing et al., 2006; 2001; McCandliss et al., 2003). These observations from the brain have inspired the development of topographic deep learning models that jointly optimize for local similarity on internal represen-

tations along with task performance (Lee et al., 2020; Margalit et al., 2024; Qian et al., 2024; Deb et al., 2025).

There are broadly three approaches to inducing topography in neural networks during training. The first involves models like TDANNs, which promote spatial topography by encouraging distance-dependent response similarity using an exponential falloff function, mimicking the spatial correlation function obtained from experimental recordings from high-level visual cortex (Lee et al., 2020; Margalit et al., 2024). The second approach relies on lateral interactions, where nearby units are encouraged to perform similar computations, as in LLCNNs and related architectures (Qian et al., 2024; Dehghani et al., 2024). The third approach, TopoNets, is inspired by early synaptic pruning and Turing patterns, promotes topography by encouraging nearby units to develop similar weights, leading to smooth tuning across a cortical sheet (Deb et al., 2025). Among the three strategies, TopoNets and TDANNs have been successfully applied to both convolutional networks and transformers, and have shown promise beyond vision, including in the language domain (Rathi et al., 2024; Deb et al., 2025; Binhuraib et al., 2025). In this work, we directly test whether topographic constraints constitute a domain-general organizing principle. Specifically, we show that topography, successfully transfers to audition, a fundamentally distinct domain that has never been tested before. This is a critical scientific test of whether the principles underpinning cortical topography are truly shared across the brain and can be leveraged for model interpretability.

## 2.2 TOPOGRAPHY IN AUDITORY CORTEX AND AUDITORY MODELS

The topographic organization of auditory responses begins in the periphery with frequency preferences, or tonotopy, of auditory nerve fibers innervating the cochlea (Dallos et al., 1996). This tonotopic structure is inherited by later brain regions subcortically and cortically (Pantev et al., 1995; Leaver & Rauschecker, 2016). Functional and anatomical subfields for other stimulus attributes have been reported in other subcortical and cortical regions, such as maps of spatial location in owl auditory midbrain Knudsen & Konishi (1978), topographical organization of responses to amplitude modulated sounds (Joris et al., 2004; Baumann et al., 2015), and specific neural populations for speech, music, and song in human auditory cortex (Norman-Haignere et al., 2015; 2022).

Tonotopic organization has long been part of the standard input representation for auditory model training, with models trained on representations such as mel-frequency cepstral coefficients (Mermelstein, 1976) and cochleagrams (Glasberg & Moore, 1990; McDermott & Simoncelli, 2011) that enforce neighboring input channels to have similar frequency tuning (Slaney, 1998). Recent years have seen a surge of brain-like deep learning models of auditory cortex (Tuckute et al., 2023) trained to perform real-world auditory tasks such as speech or music recognition, often with a biologically inspired tonotopic front-end. These models have been shown to predict human behavior and neural responses with increasing accuracy (Kell et al., 2018; Giordano et al., 2023; Tuckute et al., 2023; Francl & McDermott, 2022; Koumura et al., 2023). However, the internal topographic structure of these models has been rarely evaluated beyond investigating hierarchical organization (Kell et al., 2018; Tuckute et al., 2023), that is, early layers of the models are shown to predict primary auditory cortex, and late layers of the models are shown to predict higher-order auditory areas. Our work develops evaluation procedures for measuring the presence of smooth topographic maps in the domains of audio frequency, amplitude modulation, speech, and music.

## 3 METHODS

### 3.1 SPATIAL LOSS

To investigate how topographic constraints shape auditory representations, we adapted the TopoLoss framework (Deb et al., 2025) to the auditory domain. As before, we define a 2D "cortical sheet" from convolutional layers in the auditory model on which to enforce topography. Each convolutional kernel in the model is mapped onto this sheet. For a convolutional layer with $c_{\text{input}}$ input channels and $c_{\text{output}}$ output channels, and a kernel size of $k \times k$, the weight tensor $W \in \mathbb{R}^{c_{\text{output}} \times c_{\text{input}} \times k \times k}$ is reshaped into a cortical representation $C \in \mathbb{R}^{h \times w \times d}$, where $h \times w = c_{\text{output}}$, and $d = c_{\text{input}} \cdot k \cdot k$.

To encourage smoothness in the cortical sheet $C^{h \times w \times d}$, we apply a blurring operation that removes high-frequency variations. We compute a blurred version $C'$ of the cortical sheet using a downsampling factor $\phi_h = \phi_w = 3$ followed by upsampling:

$$\text{Blur}(X, \phi_h, \phi_w) = f_{\text{up}} \left( f_{\text{down}} \left( X, \frac{h}{\phi_h}, \frac{w}{\phi_w} \right), h, w \right) \tag{1}$$

The *TopoLoss* is then defined as the negative mean cosine similarity between the original and blurred cortical maps:

$$\mathcal{L}_{\text{topo}} = -\frac{1}{N} \sum_{i=1}^{N} \frac{C_i \cdot C_i'}{\|C_i\| \|C_i'\|} \tag{2}$$

This encourages neurons with similar functions to be spatially clustered, enhancing topographic organization. Finally, we integrate the *TopoLoss* with the primary task loss $\mathcal{L}_{\text{training}}$ as:

$$\mathcal{L}_{\text{total}} = \mathcal{L}_{\text{training}} + \tau \cdot \mathcal{L}_{\text{topo}} \tag{3}$$

where $\tau$ is a scaling coefficient controlling the influence of topographic regularization. Higher values of $\tau$ encourage stronger topographic organization.

## 3.2 TopoAudio architecture and training

**Architectures.** Our models are based on two state-of-the-art auditory neural network backbones: CochResNet50 and the Audio Spectrogram Transformer (AST). We specifically selected these two complementary architectures because they were shown to be the most accurate models of human auditory cortex responses (Tuckute et al., 2023). CochResNet50 adapts the standard ResNet50 backbone (He et al., 2015) to operate on time–frequency cochleagrams using 2D convolutions. The input to the model is a single-channel cochleagram of shape $(1, 211, 390)$, representing 211 frequency bins across 390 time steps. For the CochResNet50, the topographic loss was applied to the second convolutional layer within each residual block, promoting spatial smoothness and topographic organization across successive hierarchical stages of the network.

AST, in contrast, leverages a transformer-based architecture to model audio spectrograms through non-overlapping patches and global self-attention (Gong et al., 2021). In our adaptation, AST also operates on cochleagram, ensuring architectural differences rather than input representation drive performance differences. The topographic loss was applied to the first feed-forward projection layers within each transformer block. All models were trained on 4×H200 NVIDIA GPUs using identical multi-task objectives (see next section), ensuring a fair comparison across architectures.

**Training objective.** All TopoAudio models were trained on the Word-Speaker-Noise dataset (Feather et al., 2019), which supports multi-task learning for (1) word recognition, (2) speaker identification, and (3) background noise classification. The dataset includes 230,356 speech clips across 793 word classes and 432 speaker identities, with class sampling designed to reduce overlap (no more than 25% samples from any one word-speaker pair). Background audio was drawn from 718,625 curated AudioSet clips consisting of human and animal sounds, various musical clips, and environmental sounds to ensure diverse and high-quality noise.

Training samples included speech-only, noise-only, and speech+noise mixtures, with augmentations such as random cropping, RMS normalization, and variable SNR mixing ($-10\,\text{dB}$ to $+10\,\text{dB}$). This setup enabled supervised learning across all tasks with consistent preprocessing. This robust training procedures makes it a strong test-bed for assessing how topographic constraints influence auditory model organization and performance.

## 3.3 Model evaluations

We benchmarked the performance of models trained with and without topography across a range of auditory domains: ESC-50 (Piczak, 2015) for environmental sound classification, NSynth (Engel

et al., 2017) for musical instrument classification, and Speech Commands (Warden, 2018) for word and speaker recognition. Lastly, we evaluated these models on human auditory cortex datasets such as NH2015 (Norman-Haignere et al., 2015) and B2021 (Boebinger et al., 2021).

### 3.3.1 ACCURACY

For ESC-50, which includes 2,000 environmental sound clips across 50 categories, we followed the standard five-fold cross-validation protocol. Representations were taken from the penultimate layer of each model (e.g., `AvgPool` for ResNet50, `CLS` token for transformer). Each 5-second ESC-50 clip was randomly cropped into five 2-second segments to match the model input duration. All five crops of a training clip were used as independent training samples. During evaluation, we applied majority voting across the five crops of each test clip to determine the predicted class label. We applied cross-validation over five regularization parameters ($C = [0.01, 0.1, 1.0, 10.0, 100.0]$). The final accuracy was averaged over the five ESC-50 folds.

For NSynth, which consists of approximately 300,000 musical notes played by a variety of instruments, we focused on the task of instrument family classification. Each audio clip is 4 seconds long and labeled with one of 11 instrument families (e.g., strings, brass, keyboard). To match input requirements of our models, we used the first 2-second segment of each clip. Similar to the ESC-50 pipeline, representations were extracted from the final pooling layer of each pretrained model and used to train an SVM with similar choice of cross-validation. We adopted the official NSynth validation and test splits, training on the validation set and evaluating top-1 classification accuracy on the test set.

For Speech Commands v2, which contains approximately 100,000 1-second utterances of 35 spoken command words (e.g. yes, no, up, down). To ensure consistency with model's input during training, each audio clip was zero-padded to 2-seconds. We extracted frozen representations from the final pooling layer and trained a linear SVM to classify the command labels. We followed the standard validation-test split provided by the dataset: the SVM was trained on the validation set and evaluated on the held-out test set. Accuracy was measured as top-1 accuracy across all 35 classes.

### 3.3.2 ESTIMATING SPATIAL TOPOGRAPHY: SMOOTHING

To quantify spatial topography in model representations, we used a smoothness score that compares the tuning similarity of spatially nearby model unit pairs to that of distant pairs (Margalit et al., 2024; Deb et al., 2025). Let $x$ be a vector of pairwise tuning similarity values sorted in order of increasing cortical distance. The smoothness score $S(x)$ is then defined as:

$$S(x) = \frac{\max(x) - x_0}{x_0} \tag{4}$$

where $x_0$ is the tuning similarity for the closest unit pair, and $\max(x)$ represents the highest similarity across all distances. This metric captures how much tuning similarity drops with increasing distance, with higher values indicating smoother topographic organization across the cortical sheet. Higher smoothness indicates that units with similar representations are spatially closer, reflecting stronger topographic structure.

### 3.3.3 ESTIMATING SELECTIVITY

To visualize the selectivity of model units for specific categories, we extracted layer-wise representations in response to annotated stimuli and applied a standard t-statistic-based measure. Specifically, for each target category $c$, we compared the distribution of activation values to those from other categories $o$ using the following formula:

$$t = \frac{\mu_c - \mu_o}{\sqrt{\frac{\sigma_c^2}{N_c} + \frac{\sigma_o^2}{N_o}}} \tag{5}$$

Here, $\mu$, $\sigma$, and $N$ represent the mean, standard deviation, and number of samples for each category, respectively. This score reflects how strongly a unit differentiates the target category from others,

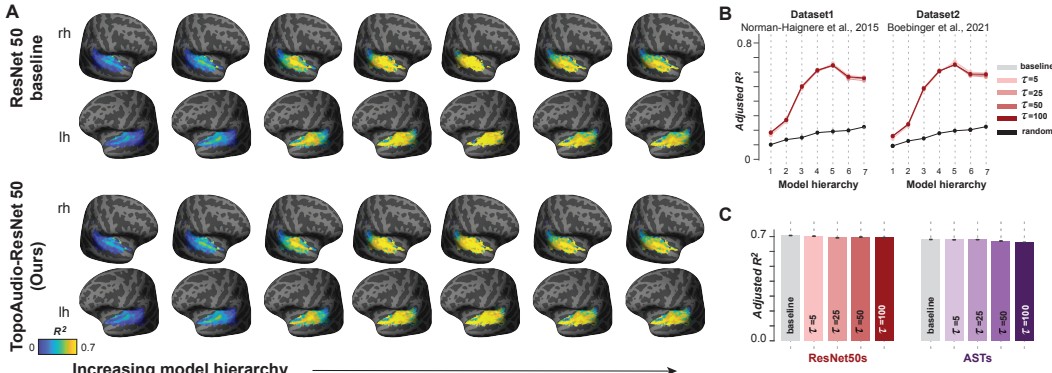

Figure 2: **Topographic auditory models maintain overall voxel-wise brain predictivity.** A) Brain maps display peak voxel-wise predictions across the auditory cortex for the baseline and topographic ResNet50. The colormap indicates the mean variance explained at each cortical vertex, averaged across subjects on the fsaverage surface. The model layers are shown from left to right, corresponding to increasing depth in the network hierarchy. B) ResNet50 model prediction accuracy as a function of depth in the network hierarchy for variants of the ResNet50 model on the NH2015 (Norman-Haignere et al., 2015) and B2021 (Boebinger et al., 2021) datasets. Note that the baseline model and the TopoAudio are highly overlapping. C) Bar plot summarizing peak encoding performance across all voxels for each model variant, including the random-initialized, baseline, and TopoAudio models for both the ResNet50 and AST architectures.

with higher values indicating greater selectivity. Layer maps are visualized by showing selectivity measured from each filter (for convolution-based models) or units (for transformer-based models) arranged on the 2D cortical sheet. When units with similar preferences cluster together spatially, the map appears smooth or patchy demonstrating topographic structure for the measured feature; when preferences are scattered, the map looks disorganized Figures 3 and 4.

### 3.3.4 FEATURES AND VOXEL DECOMPOSITION

Functional selectivities for speech and music are often not directly observable at the voxel level in fMRI data, and are instead inferred by decomposing voxel responses into a small number of latent components Norman-Haignere et al. (2015). To analyze the structure of model and brain responses, we adapted this voxel decomposition technique to the fMRI response matrix and model activations. Specifically, we adopted Non-parametric decomposition algorithm (Norman-Haignere et al., 2015), which factorizes a data matrix $D \in \mathbb{R}^{S \times V}$ (where $S$ for stimuli and $V$ for voxels/features) into a response matrix $R \in \mathbb{R}^{S \times N}$ and a weight matrix $W \in \mathbb{R}^{N \times V}$ (where $N$ for number of components), such that:

$$D \approx RW \tag{6}$$

This method is conceptually related to Independent Component Analysis (ICA), but differs in its approach to estimating non-Gaussianity. Instead of using contrast functions or kurtosis, it directly minimizes the entropy of the weight distribution using a histogram-based entropy estimator. This allows for the discovery of meaningful structure in high-dimensional data, including sparse or skewed distributions (Figure 4). This enabled us to extract shared components that capture interpretable spatial structure - such as tonotopic and category-selective regions. As has been shown in prior work, 6-components were sufficient to account for more than 80% of the noise-corrected variance in voxel responses (Norman-Haignere et al., 2015). Similarly, we adopted $N = 6$ components for features and voxel decomposition.

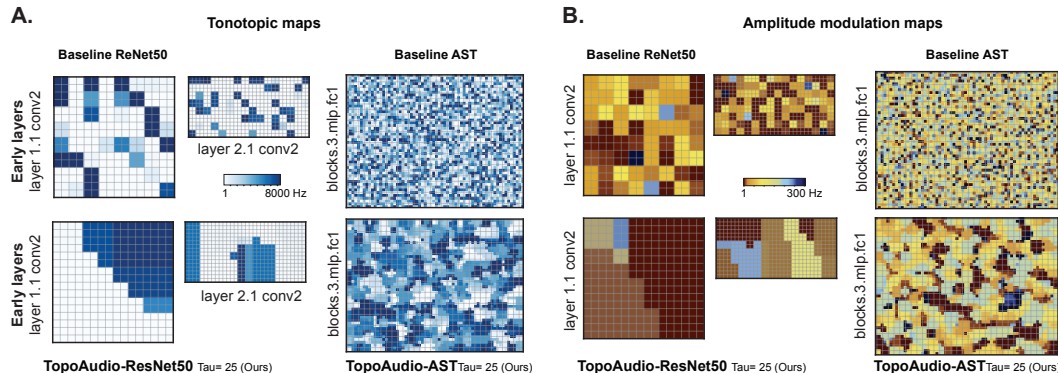

Figure 3: **TopoAudio models exhibit signatures of early auditory representations.** A) Tonotopic maps in baseline ResNet50s and ASTs (top row) are spatially disorganized, whereas TopoAudio models with topographic constraints ($\tau = 25$ for ResNet50s, $\tau = 5$ for ASTs; bottom row) exhibit smooth, spatially clustered frequency maps. B) Amplitude modulation tuning maps show a similar pattern: baseline models lack clear organization (top row), while topographic models (bottom row) develop coherent spatial clusters resembling auditory cortical maps.

## 4 RESULTS

### 4.1 TOPOAUDIO MODELS PRESERVE PERFORMANCE UNDER TOPOGRAPHIC CONSTRAINTS

A typical concern with topographic neural network models is that introducing spatial constraints often leads to substantial drops in task performance on standard engineering tests (Qian et al., 2024). To examine whether this tradeoff holds in the auditory domain, we evaluated TopoAudio with varying levels of topographic smoothness (controlled by hyperparameter $\tau$), against baseline non-topographic models on three benchmark datasets: environmental sounds, speech, and music (see section 3).

Appendix Table 1 shows the model performance accuracy for all models across each evaluation domain. Critically, despite the additional topographic constraint, all TopoAudio variants maintained high classification accuracy across tasks. Performance was consistently comparable to the non-topographic baseline model. These results demonstrate that topographic organization can be induced in auditory networks without sacrificing task performance, and establish TopoAudios as competitive auditory models suitable for further analyses.

### 4.2 TOPOAUDIOS PREDICT HUMAN FMRI RESPONSES WITH HIGH ACCURACY

A key requirement for any biologically inspired model is that it must retain strong predictive performance while better aligning with the structure of neural data. To assess whether brain predictivity is maintained even with topographic constraints ($\tau$), we evaluated the predictive abilities of TopoAudios and the baseline ResNet50 architecture using two fMRI datasets, NH2015 (Norman-Haignere et al., 2015) and B2021 (Boebinger et al., 2021). Following prior work (Tuckute et al., 2023), we computed the voxel-wise explained variance (adjusted $R^2$) using linear regression across each layer of the model hierarchy.

Figure 2 shows that both baseline and TopoAudios achieved comparable levels of predictivity across layers. This is evident in the qualitative prediction maps projected onto the cortical surface (Figure 2A), where both models showed similar levels of adjusted $R^2$ across voxels. Both models also showed characteristic features of cortical predictivity. Prediction accuracies improve across model layers. This is apparent more clearly in Figure 2B which shows that model prediction accuracy peaks around the 6th topographic layer. It is particularly striking that prediction accuracy is nearly indistinguishable between the baseline and TopoAudios. The five curves corresponding to the baseline model and the different topographic variants of TopoAudios are virtually superimposed. As a further quantification, Figure 2C demonstrates that adding the topographic constraint does little to reduce the overall model prediction accuracy in both the ResNet50 architecture and the AST archi-

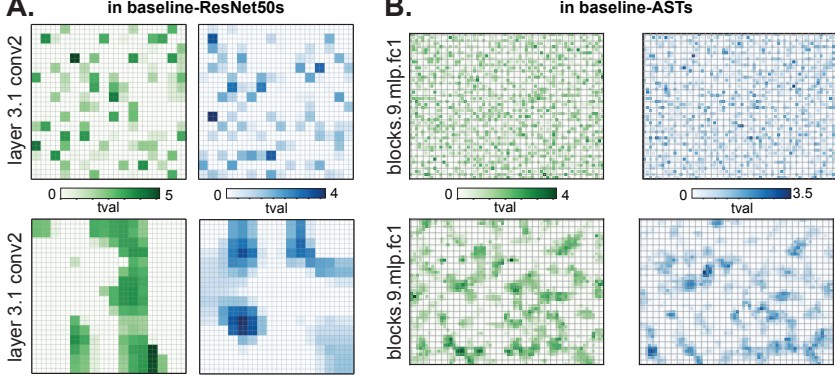

Figure 4: **TopoAudio models show late auditory representations related to speech and music selectivity.** Selectivity maps show $t$-values for speech (green) and music (blue) preferences, with baseline models (top row) lacking the clear modular structure seen in the TopoAudio models (bottom row).

tecture. This pattern held not only across the whole auditory cortex but also within specific regions of interest (ROIs obtained from (Norman-Haignere et al., 2015)). When we repeated the analysis separately for early tonotopic regions, pitch-selective regions, and higher-order music and speech areas, we observed the same trends. TopoAudios were as predictive as baseline models for all ROIs. Together these results confirm that inducing topographic constraints do not impair their ability to predict fMRI responses in auditory cortex. Thus, TopoAudios preserve both task performance (on engineering metrics) and predictive accuracy (on 2 distinct fMRI datasets) while producing spatially structured internal representations. The similarity of the TopoAudios to the baseline models leads to a natural question: **are these models actually the same, or are our evaluations not sensitive enough to reveal the differences?**

### 4.3 TOPONETS RECAPITULATE THE SIGNATURES OF EARLY AND LATE AUDITORY PROCESSING IN THE BRAIN

To evaluate whether TopoAudios develop biologically meaningful auditory representations, we examined whether these models show known signatures of auditory processing. Specifically, we assessed whether the topographic structures within the TopoAudios give rise to tonotopic and amplitude modulation (AM) maps, typically considered features of early auditory processing, as well as selective responses to music and speech, characteristic of higher-order auditory areas.

Figure 3A shows the observed tonotopic maps in the early layers (layer 1.1 and layer 2.1 in ResNet50s and blocks.9.mlp.fc1 in ASTs) of the baseline and topographic models. The frequency preferences appear randomly distributed on the cortical sheet for the baseline models (top). In contrast, the same layers for the TopoAudio model (below) show a smooth frequency gradients which qualitatively resembles the tonotopic organization observed in primary auditory cortex. Similarly, in the AM tuning maps (Figure 3B), the baseline model is fragmented and disorganized, while the TopoAudio model develops smooth spatial transitions, considered another hallmark of early auditory areas.

Figure 4 examines the late-stage selectivity for speech and music observed in the human auditory cortex using fMRI-like selectivity maps (see section 3) applied to a later layer (layer 3.1 in ResNet50 and blocks.9.mlp.fc1 in ASTs) of baseline and TopoAudio models. For both speech (green) and music (blue), baseline models (top row) show spatially inconsistent selectivity on the cortical sheet. By constrast, TopoAudios exhibit spatially clustered patches of selectivity for speech and music, mirroring the modular organization observed in non-primary auditory cortex. This result is striking, as such modularity for music and speech emerged purely from task optimization with spatial con-

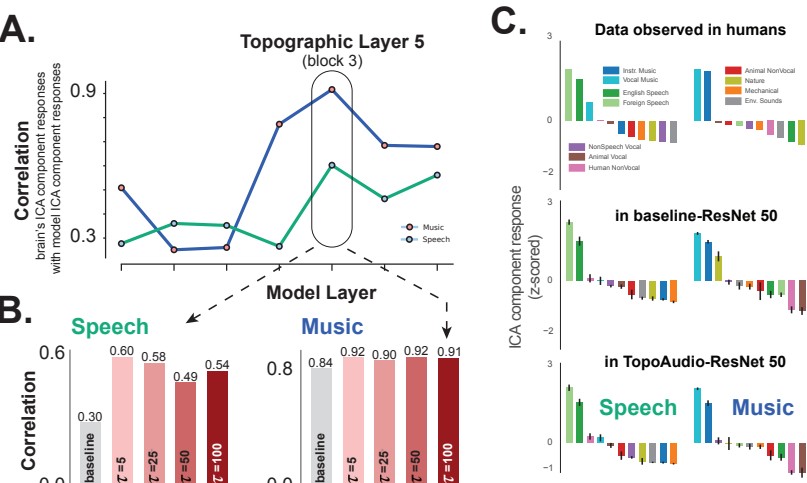

Figure 5: **TopoAudio models better align with human speech and music-selective cortical components.**(A) Correlation between model-derived ICA components and brain-inferred ICA components selective for speech (green) and music (blue), across layers of the ResNet50 hierarchy. TopoAudios show the highest similarity with brain components at layer 3, consistent with the emergence of high-level selectivity in the auditory cortex.(B) Summary of brain-model ICA component correlations for each TopoNet variant ($\tau = 5$–$100$) compared to the baseline ResNet50. Topographic models yield higher median correlation with brain-derived components for both music and speech. (C) Category response profiles of ICA componenets derived from human fMRI (top), baseline ResNet50 (middle), and TopoAudio (bottom). Responses are grouped by stimulus categories.

straints. However, a limitation of this finding is its qualitative nature. To address this concern, we next quantified the correspondence between model-based selectivity maps and empirical fMRI data from human auditory cortex

Using human fMRI responses to a large set of natural sounds, selectivity for music and speech is often inferred by factorizing the fMRI responses from the auditory cortex into spatial components using ICA. This approach consistently reveals components that are selectively responsive to music or speech stimuli (Norman-Haignere et al., 2015; Boebinger et al., 2021). To evaluate whether such category-selective structure emerges in our models, we applied the same analysis to the activation patterns in TopoAudios. For each model layer, we performed the non-parameteric decomposition algorithm (Norman-Haignere et al., 2015) on the model unit responses (using six components, consistent with human analyses) and identified the component that best matched the brain-inferred music or speech component. In Figure 5A, we show the correlation between brain and model components across the model hierarchy. For both music and speech, TopoAudios showed the highest correlation with the fMRI components at layer 3, mirroring the depth at which late-stage selectivity emerges in the human auditory cortex.

We next quantified these correlations across all model variants. As shown in Figure 5B, the TopoAudios consistently produced higher correlations with brain-inferred components than the baseline models. For the music component, the median correlation across TopoAudio variants was 0.915, compared to 0.84 for the baseline. For the speech component, the TopoAudio models again showed higher median correlation (0.56) compared to the baseline (0.30). These results indicate that topographic constraints enhance the emergence of music- and speech-selective representations that align more closely with those observed in the human brain. We then plotted the response profiles of the components derived from humans, the baseline model, and the TopoAudio models grouped by the auditory stimulus categories from the original dataset (Norman-Haignere et al., 2015) in Figure 5C. The inferred components from the brain (top), the baseline model (middle) and the TopoAudios exhibited clear and selective responses to music and speech categories, confirming that the model components capture aspects of music and speech selectivity also observed human brains.

## 5 DISCUSSION

In this work, we tested whether the topographic organization of the auditory cortex could emerge in deep neural networks by optimizing for auditory tasks under a spatial smoothness constraint. We introduced TopoAudio, the first end-to-end topographic models of the auditory system. These models recapitulate the hierarchical organization of auditory cortex and reproduce hallmark features such as smooth tonotopic and amplitude-modulation maps in early layers, and spatially clustered selectivity for music and speech in later layers, all while maintaining strong auditory task performance and fMRI predictivity. None of these features were hand-engineered. They emerged naturally from task-driven optimization combined with a topographic constraint. Crucially, the topographic models also provide a closer match to the internal structure of auditory cortex. Compared to non-topographic baselines, TopoAudio models align significantly better with ICA maps derived from human fMRI data. This suggests that our models not only predict the brain's responses but also better capture the fine-grained representational geometry that underlies how the brain organizes sound. In doing so, they offer a stronger test of model–brain similarity, moving beyond surface-level prediction toward an assessment of deeper structural and computational alignment.

Our findings in audition, combined with prior topographic modeling work in vision and language, provide strong corroborating evidence for a unified computational strategy employed by the brain to efficiently encode information under its architectural constraints. This success in a new sensory domain provides a powerful test of the principle's generality, showing it is not a quirk of visual processing but a foundational component of sensory representation across the cortex. Further, the similarity of topographic structure between models and biological systems can serve as an additional evaluation of models in an era where many models seem to be performing equally well on many of our standard benchmarks (Tuckute et al., 2023; Conwell et al., 2024; Ratan Murty et al., 2021; Feather et al., 2025).

The primary focus of this work was to evaluate whether our topographic auditory models capture the representational features observed in biological auditory systems. It is important to note that there is a long history in auditory neuroscience of searching for auditory maps. Multiple studies have suggested that the topographic structure of auditory cortex might in fact be quite complex (Schreiner & Winer, 2007; Kanold et al., 2014; Middlebrooks, 2021), and that we do not yet have a good description of its organization. For example, even though there is general agreement that speech and music selective neural populations exist in the brain, the general locations and numbers of such patches are not well characterized. Future data collected to characterize the biological topography of the auditory system can be incorporated into evaluations of topographic models. Additionally, by building topographic models of auditory cortex trained on natural stimuli, TopoAudios can serve as a computational tool for exploring and testing theories of cortical organization, potentially revealing previously unknown structure in auditory cortex.

**Limitations and Future work:** This work serves as a starting point for topographic auditory models. Here, we only investigated two DNN architectures trained on the same task, but the same type of topographic loss could be applied to other DNN architectures or combined with a different set of tasks. Additionally, other datasets may be better for evaluating details of the topographic structure of the auditory system, as the fMRI dataset lacks fine-grained temporal resolution and may hide some spatial structure Norman-Haignere et al. (2022). Nevertheless, the present results provide a computational framework for studying the emergence of cortical organization toward models that unify task performance with biological structure.

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

# A APPENDIX

## A.1 TOPOAUDIO PERFORMANCE

Table 1 reports classification accuracy and representational smoothness for both baseline and topographic variants of Transformer-B/32 and ResNet-50 backbones. Across datasets (ESC50, NSynth, and Speech Commands), accuracy remains nearly unchanged when introducing topographic constraints ($\tau$), with performance differences typically within $< 1\%$ of baseline. In contrast, smoothness values increase substantially, confirming that topographic regularization induces more spatially coherent representations. These results demonstrate that TopoAudio models preserve strong classification performance while simultaneously improving internal topographic structure, supporting their utility as both effective and interpretable auditory models.

Table 1: **Topographic auditory models maintain high classification performance across evaluations.** Accuracy is reported for ESC50, NSynth, and Speech Command datasets using Transformer-B/32 and ResNet-50 backbones. While baseline models achieve slightly higher accuracy, introducing topographic constraints ($\tau$) substantially increases representational smoothness with only modest changes in classification performance. Topographic Avg. indicates the mean performance across all non-baseline $\tau$ values.

| Topography ($\tau$) | Accuracy | | | Smoothness |
|---|---|---|---|---|
| | ESC50 | NSynth | SpeechCmd | |
| **Transformer-B/32** | | | | |
| Baseline | 82.10 | 98.25 | 92.94 | 0.31 |
| 5 | 81.94 | 98.13 | 92.63 | 0.46 |
| 25 | 82.01 | 98.13 | 92.80 | 0.50 |
| 50 | 81.66 | 97.99 | 92.42 | 0.57 |
| 100 | 81.88 | 97.91 | 92.38 | 0.56 |
| *Topographic Avg.* | 81.87 | 98.04 | 92.56 | 0.52 |
| **ResNet-50** | | | | |
| Baseline | 81.69 | 98.29 | 86.68 | 0.34 |
| 5 | 81.46 | 98.50 | 86.88 | 1.10 |
| 25 | 80.62 | 98.39 | 87.89 | 0.87 |
| 50 | 80.84 | 98.36 | 87.57 | 1.04 |
| 100 | 80.32 | 98.72 | 86.85 | 1.09 |
| 200 | 80.70 | 98.48 | 87.47 | 0.93 |
| *Topographic Avg.* | 80.79 | 98.49 | 87.33 | 1.01 |

# B COMPUTATION OF MORAN'S I SPATIAL AUTOCORRELATION

Moran's I is a metric of global spatial autocorrelation that quantifies the degree to which values defined over a spatial domain exhibit smooth clustering or random dispersion (Rathi et al., 2024). Formally, for a set of values $x_i$ arranged over $N$ spatial units, Moran's I is defined as:

$$I = \frac{N}{W} \frac{\sum_i \sum_j w_{ij}(x_i - \bar{x})(x_j - \bar{x})}{\sum_i (x_i - \bar{x})^2}, \tag{7}$$

where $\bar{x}$ is the global mean, $w_{ij}$ are entries of a sparse adjacency matrix encoding the neighborhood relationships among vertices or units, and $W = \sum_{i,j} w_{ij}$ is the total connection weight. Values of Moran's I range from $-1$ to 1, with positive values reflecting spatially smooth and contiguous organization, values near zero indicating spatial randomness, and negative values indicating systematic spatial dispersion.

### B.1 MORAN'S I FOR BRAIN MAPS

To compute Moran's I for human auditory cortex, we generated several cortical maps derived from fMRI responses, including music-selective and speech-selective contrast maps. We constructed a binary vertex adjacency matrix in which each vertex is connected to its immediate geometric neighbors in the surface tessellation (typically six per vertex). Prior to computing Moran's I, vertices failing to meet a statistical significance threshold (e.g., FDR-corrected $p < 0.05$) were zero-masked to remove noise-driven fluctuations. Moran's I was then evaluated directly on these spatial contrast maps, yielding a quantitative estimate of the smoothness of the underlying functional organization on category-selective gradients.

### B.2 MORAN'S I FOR COMPUTATIONAL MODELS

For neural network models, we applied an analogous spatial autocorrelation procedure by imposing a fixed two-dimensional topographic layout over the units of each layer. Specifically, units within each MLP block (or convolutional feature map) were assigned to positions on a 2D grid, and a binary adjacency matrix was constructed based on local grid connectivity. Using this spatial structure, we computed selectivity maps for each model, including frequency tonotopy maps, modulation-rate maps, and music- vs.-other and speech- vs.-other contrast maps derived from model activations. Moran's I was computed for each layer individually and then averaged across relevant layers to obtain a model-level smoothness estimate for each topographic strength $\tau$. This procedure mirrors the approach used for fMRI maps, enabling direct comparison between cortical organization and the representational smoothness learned by neural networks trained with and without topographic constraints.

### B.3 MORAN'S I RESULTS

Table 2 summarizes the Moran's I values for the two model classes considered—ResNet-50 and AST-Base—across varying topographic strengths $\tau$. Several clear patterns emerge from this comparison. First, the baseline condition ($\tau = 0$) produces Moran's I values at or near zero across all four map types (tonotopy, amplitude modulation, music, and speech), confirming that without explicit spatial regularization neither architecture develops meaningful topographic structure. This reinforces that standard convolutional or transformer-based audio models do not spontaneously acquire spatial contiguity in their feature maps.

Introducing topographic regularization ($\tau > 0$), however, yields large and systematic increases in spatial autocorrelation for both architectures. In AST-Base, smoothness increases consistently from $\tau = 5$ to $\tau = 50$, with only a slight saturation at $\tau = 100$, indicating that moderate levels of spatial pressure are sufficient to induce stable and biologically interpretable organization. Similar improvements appear in ResNet-50, where $\tau = 50$ and $\tau = 100$ produce substantial topographic structure across all selectivity types. Notably, music- and speech-selective maps in ResNet-50 reach Moran's I values in the 0.80–0.83 range, approaching the upper-bound smoothness observed in human auditory cortex.

Finally, the brain exhibits near-ceiling spatial autocorrelation (0.99 for both music and speech), serving as a reference for the maximum smoothness achievable in biological systems. The fact that topographically constrained models—particularly at higher $\tau$ levels—approach these values demonstrates that imposing spatial continuity enables artificial networks to replicate key hallmarks of cortical organization. This alignment between model- and brain-derived maps highlights the utility of topographic constraints as a principled mechanism for shaping neural network representations toward neurobiological structure.

Table 2: **Topographic smoothness of music-, speech-, tonotopic-, and AM-selective maps.** Moran's I spatial autocorrelation for ResNet-50, AST-Base, and human brain fMRI.

| Selectivity | ResNet-50 | | | | | AST-Base | | | | | Brain (NH2015) |
|---|---|---|---|---|---|---|---|---|---|---|---|
| | $\tau=0$ | $\tau=5$ | $\tau=25$ | $\tau=50$ | $\tau=100$ | $\tau=0$ | $\tau=5$ | $\tau=25$ | $\tau=50$ | $\tau=100$ | |
| **Tonotopy** | -0.01 | 0.54 | 0.59 | 0.67 | 0.67 | 0.00 | 0.43 | 0.50 | 0.52 | 0.49 | – |
| **Amplitude Modulation** | -0.01 | 0.55 | 0.56 | 0.34 | 0.46 | 0.00 | 0.33 | 0.38 | 0.45 | 0.44 | – |
| **Music** | 0.01 | 0.79 | 0.78 | 0.83 | 0.80 | 0.00 | 0.62 | 0.71 | 0.73 | 0.73 | 0.99 |
| **Speech** | -0.01 | 0.75 | 0.78 | 0.79 | 0.82 | 0.00 | 0.61 | 0.69 | 0.71 | 0.72 | 0.99 |

# C  FMRI DATASETS

## C.1  NH2015

The fMRI data used in this study are a subset of those originally reported in (Norman-Haignere et al., 2015), with procedures summarized below.

**Participants and Experimental Design.**  Eight right-handed, native English-speaking participants (4 female; mean age 22 years, range 19–25) with normal hearing and no formal musical training participated in the study. Each participant completed three fMRI sessions (~2 hours each). Five additional participants were excluded due to either incomplete scanning sessions or excessive head motion and task non-compliance. All participants gave informed consent under protocols approved by the MIT Committee on the Use of Humans as Experimental Subjects (protocol 2105000382).

**Stimuli.**  A total of 165 two-second natural sounds were selected to span a wide range of real-world auditory categories. Each sound was validated using a 10-way forced-choice classification task on Amazon Mechanical Turk and included only if recognized with at least 80% accuracy. Stimulus names and categories are available in the supplementary materials of (Tuckute et al., 2023), and the full stimulus set can be downloaded from: `http://mcdermottlab.mit.edu/downloads.html`.

**fMRI Procedure.**  Stimuli were presented in a blocked design, with each block consisting of five repetitions of the same 2-second sound, interleaved with 200 ms of silence to minimize scanner noise. Each block lasted 17 s (TR = 3.4 s), and silence blocks of equal duration were interspersed to estimate baseline responses. To ensure attentiveness, participants performed an intensity discrimination task in each block, identifying the quietest sound (7 dB lower than the others) via button press.

**Data Acquisition.**  Data were acquired on a 3T Siemens Trio scanner with a 32-channel head coil at the Athinoula A. Martinos Imaging Center (MIT). Each run consisted of 15 slices oriented parallel to the superior temporal plane (TR = 3.4 s, TE = 30 ms, flip angle = 90°). The in-plane resolution was 2.1 mm × 2.1 mm, with 4 mm thick slices and a 10% gap (voxel size: $2.1 \times 2.1 \times 4.4$ mm). The first 5 volumes of each run were discarded.

**Preprocessing.**  Preprocessing was conducted using FSL, FreeSurfer, and custom MATLAB scripts. Functional data were motion- and slice-time corrected, linearly detrended, skull-stripped, and aligned to each participant's anatomical scan using FLIRT and BBRegister. Volumes were projected to the reconstructed cortical surface using FreeSurfer and smoothed with a 3-mm FWHM 2D Gaussian kernel. Percent signal change was computed relative to silence blocks, and responses were downsampled to a 2-mm isotropic grid on the FreeSurfer surface. All participants' data were registered to the `fsaverage` template.

**Voxel Selection.**  Voxel selection followed the criteria in (Tuckute et al., 2023). We retained voxels within a superior temporal and posterior parietal mask if they met two conditions: (1) significant sound vs. silence response ($p < 0.001$, uncorrected), and (2) reliable responses to sounds across scan sessions, quantified as:

$$r = 1 - \frac{\|\mathbf{v}_{12} - \text{proj}_{\mathbf{v}_3} \mathbf{v}_{12}\|_2}{\|\mathbf{v}_{12}\|_2}, \quad \text{with} \quad \text{proj}_{\mathbf{v}_3} \mathbf{v}_{12} = \left(\frac{\mathbf{v}_3 \cdot \mathbf{v}_{12}}{\|\mathbf{v}_3\|_2^2}\right) \mathbf{v}_3$$

Here, $\mathbf{v}_{12}$ is the voxel's response vector (averaged over the first two sessions) to all 165 sounds, and $\mathbf{v}_3$ is the same voxel's response from the third session. This measure captures the fraction of variance in $\mathbf{v}_{12}$ explained by $\mathbf{v}_3$. Voxels with $r \geq 0.3$ were retained. Across participants, this yielded 7,694 voxels (mean per participant: 961.75; range: 637–1,221).

### C.2   B2021

The B2021 fMRI dataset used in this study was originally collected and analyzed by (Boebinger et al., 2021) and reanalyzed in (Tuckute et al., 2023). We summarize the methodological details below.

**Participants and Experimental Design.** Twenty right-handed participants (14 female; mean age: 25 years, range: 18–34) each completed three fMRI sessions (~2 hours per session). Half of the participants ($n = 10$) were highly trained musicians, with an average of 16.3 years (SD = 2.5) of formal training that began before age 7 and continued through the time of scanning. The other half ($n = 10$) were non-musicians with fewer than 2 years of musical training, none of which occurred before age 7 or within 5 years of scanning. All participants provided informed consent, and the study was approved by the MIT Committee on the Use of Humans as Experimental Subjects (protocol number 2105000382).

**Stimuli.** The stimulus set consisted of 192 natural sounds, including 165 from (Norman-Haignere et al., 2015) and 27 additional music and drumming clips representing diverse musical cultures. To ensure comparability with NH2015, all analyses in this study were restricted to the shared subset of 165 sounds.

**fMRI Procedure.** The scanning procedure closely followed that of (Norman-Haignere et al., 2015), with some modifications. Each stimulus block consisted of three repetitions of a 2-second sound, lasting 10.2 seconds total (TR = 3.4 s, 3 repetitions). Each participant completed 48 runs across the 3 sessions (16 runs per session), with each run containing 24 stimulus blocks and 5 randomly interleaved silent blocks. This design enabled each sound block to be repeated 6 times across the experiment. Participants performed an intensity discrimination task, pressing a button upon detecting the quietest of the three repetitions in a block (12 dB lower).

**Data Acquisition.** MRI data were collected using a 3T Siemens Prisma scanner with a 32-channel head coil at the Athinoula A. Martinos Imaging Center at MIT. Functional volumes (48 slices per volume) covered the superior temporal and parietal lobes, matching the anatomical mask used in (Norman-Haignere et al., 2015). Imaging parameters were: TR = 3.4 s (TA = 1 s), TE = 33 ms, flip angle = 90°, in-plane resolution = 2.1 mm, slice thickness = 3 mm (10% gap), and voxel size = $2.1 \times 2.1 \times 3.3$ mm. A multiband SMS factor of 4 was used to accelerate acquisition. Structural T1 images (1 mm isotropic) were also collected.

**Preprocessing.** Preprocessing matched the pipeline used in (Norman-Haignere et al., 2015), but with a general linear model used to estimate voxel responses due to the shorter stimulus blocks and increased overlap in BOLD responses. For each stimulus block, beta weights were computed using a boxcar function convolved with a canonical hemodynamic response function, along with 6 motion regressors and a linear trend term. Resulting beta weights were downsampled to a 2-mm isotropic grid on the FreeSurfer cortical surface. Each participant's cortical surface was registered to the `fsaverage` template.

**Voxel Selection.** Voxels were selected using the same reliability-based procedure described in (Tuckute et al., 2023). Reliability was computed from vectors of beta weights for the 165 shared stimuli, estimated separately from two halves of the data ($v_1$ = runs 1–24, $v_2$ = runs 25–48):

$$r = 1 - \frac{\|\mathbf{v}_{12} - \mathrm{proj}_{\mathbf{v}_3}\mathbf{v}_{12}\|_2^2}{\|\mathbf{v}_{12}\|_2^2}, \quad \text{where} \quad \mathrm{proj}_{\mathbf{v}_3}\mathbf{v}_{12} = \left(\frac{\mathbf{v}_3 \cdot \mathbf{v}_{12}}{\|\mathbf{v}_3\|_2^2}\right)\mathbf{v}_3$$

Voxels with $r \geq 0.3$ and significant sound-evoked responses ($p < 0.001$, uncorrected) were retained. This procedure yielded a total of 26,792 reliable voxels across 20 participants (mean: 1,340 per participant; range: 1,020–1,828).

### C.3 Voxelwise Response Modeling

This procedure was repeated 10 times (once per train-test split), and the median corrected variance explained was reported for each voxel-layer pair. We evaluated all layers from each candidate model on both datasets, yielding voxelwise explained variance values for 7,694 voxels (NH2015) and 26,792 voxels (B2021).

**Regularized linear regression and cross-validation.** To model the relationship between model unit activations and measured brain responses, we used voxelwise linear encoding models. For each voxel, we predicted its time-averaged response to natural sounds as a linear combination of time-averaged activations from a specific model layer. We randomly split the 165 sounds into 10 unique train-test partitions of 83 training and 82 test sounds. For each split, we fit a regularized linear regression (ridge regression) model using the 83 training sounds and evaluated prediction performance on the held-out 82 sounds.

**Regression formulation.** Let $\mathbf{y} \in \mathbb{R}^n$ be the voxel's mean response to $n = 83$ sounds, and let $\mathbf{X} \in \mathbb{R}^{n \times d}$ be the matrix of $d$ regressors (i.e., time-averaged activations from a model layer). The ridge solution is:

$$\mathbf{w} = (\mathbf{X}^\top\mathbf{X} + n\lambda\mathbf{I})^{-1}\mathbf{X}^\top\mathbf{y}$$

where $\lambda$ is the regularization parameter and $\mathbf{w}$ is the vector of regression weights. Both $\mathbf{y}$ and the columns of $\mathbf{X}$ were demeaned (but not normalized) prior to regression. This allowed units with greater magnitude variance to contribute more to the prediction under a non-isotropic Gaussian prior. To avoid data leakage, all transformations were learned on the training set and applied to the test set.

We used leave-one-out cross-validation within the 83 training sounds to select $\lambda$. For each of 100 logarithmically spaced values (from $10^{-50}$ to $10^{49}$), we computed the mean squared error of the predicted response for each left-out training sound. The $\lambda$ minimizing this error was used to retrain the model on all 83 training sounds. The final model was then used to predict responses to the 82 held-out test sounds, and performance was quantified using the Pearson correlation between predicted and actual voxel responses. Negative correlations or correlations with zero variance were set to zero.

**Correcting for reliability of predicted voxel responses.** Because both training and test responses are affected by measurement noise, we corrected for the reliability of both the predicted and measured voxel responses. This correction was essential to fairly compare model performance across voxels and model layers. We defined the corrected variance explained using the attenuation-corrected squared correlation:

$$r_{\mathbf{v},\hat{\mathbf{v}}}^{2*} = \frac{r(\mathbf{v}_{123}, \hat{\mathbf{v}}_{123})^2}{r_v' r_{\hat{v}}'}$$

where $\mathbf{v}_{123}$ is the voxel response to the 82 test sounds, $\hat{\mathbf{v}}_{123}$ is the predicted response, and $r_v'$, $r_{\hat{v}}'$ are the reliabilities of the measured and predicted responses, respectively. Reliability was estimated via median Spearman–Brown corrected correlations across scan pairs. For stability, we excluded voxels for which $r_v'$ or $r_{\hat{v}}'$ was less than $k = 0.182$ and $k = 0.183$, respectively (corresponding to $p < 0.05$ thresholds for 83- and 82-dimensional Gaussian variables).

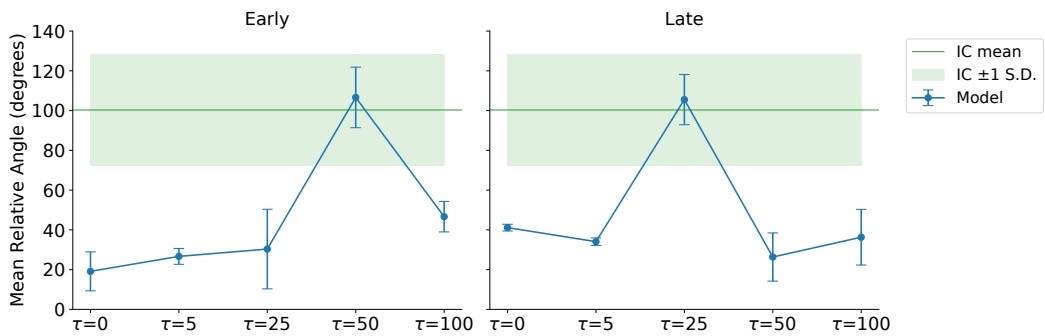

Figure 6: A stronger topographic constraint yields more perpendicular gradients for tonotopy and modulation frequency until $\tau = 50, 25$ for early and late layers respectively.

## D ANALYSING PERPENDICULARITY OF GRADIENTS FOR TONOTOPY VERSUS MODULATION RATE

It is known from Baumann et al. (2011) and Baumann et al. (2015) that the gradient of modulation rate and tonotopy run perpendicularly to each other in the auditory cortex. To check for this property in our topgraphic audio models, we adopted the method specified in the supplementary material of Baumann et al. (2011) which involves performing a planar regression on each 2D preference map, extracting the direction of the steepest gradient from the fitted plane, and computing the relative angle between the tonotopic and AM-rate gradients. For both early and late layers, we observe that a stronger topographic constraint yields more perpendicular gradients – and hence a better match with the actual data from the Inferior Colliculus (IC) (marked in green as IC mean and IC std in Figure 6). We also observe that after a certain threshold, the angles between the gradients start decreasing again.

**LLM usage in work.** LLMs were used for editing grammar, spelling, and suggesting revisions for clarity in the writing.

