# OpenReview forum: "End-to-end topographic auditory models replicate signatures of human auditory cortex"
_ICLR.cc/2026/Conference — Submitted to ICLR 2026_

### Official Review · Reviewer_UF17 · 2025-10-29

**Soundness:** 2
**Presentation:** 2
**Contribution:** 3
**Rating:** 2
**Confidence:** 5

**Summary:**

The authors used two topographical models to replicate the spatial organizations in the human auditory cortex, including tonotopy (spectral frequency), amplitude-modulation (temporal frequency) maps, and patches for speech and music. They employed the same model previously used in the topographic vision model (Deb et al., 2025, ICLR).

**Strengths:**

Originality:

This is the first end-to-end model that tries to model the spatial organization (maps and patches) in the human auditory cortex.

Quality:

See “Weaknesses” and “Questions.”

Clarity:

The manuscript is easy to understand, and the figures are well illustrated. However, some parts are confusing. See “Weaknesses” and “Questions.”

Significance:

We need a computational model to explain the organization of the auditory cortex. This work will make a big contribution to our understanding of the auditory cortex and has broad implications for other sensory systems as well.

**Weaknesses:**

**First**, this is not a method paper that proposed a new loss function. The authors used the same method as TopoNets (Deb et al., 2025, ICLR).

**Second**, since this is not a method paper, the contribution should be solely in the modeling or interpretation of neuroscience findings. I do not agree with the authors’ claim that “TopoAudio is the first end-to-end biologically grounded auditory model to exhibit emergent topography” (L29–L30). It only shows certain kinds of patches, but no clear topography. In L409 and L412, the authors mention “a smooth frequency gradient” and “smooth spatial transitions.” I do not think their results are smooth or show gradients.

At the bottom of Figure 3, the ResNet-50 model shows one or two patches for spectral frequency and four or seven patches for temporal-modulation frequency. In contrast, the AST models show numerous (over 30?) much smaller patches. This problem may result from the extremely low spatial resolution in the ResNet-50 model. Why can’t the authors use a higher resolution, as in the TDANN paper (Figure 2C, right)?

There is no gradient or transition in Figure 3B (temporal frequency or AM map). In the ResNet-50 model, the changes are arbitrary from one frequency to another. I do not know how to interpret the results from the AST model; it looks like a blurred version of the baseline model. Please correct me if I am wrong. In Figure 3A (spectral frequency or tonotopy), the frequency change in layer 2.1 is random, though layer 1.1 does exhibit a three-color gradient. Is this the topography the authors refer to? They need to show more evidence to convince us there is a gradient in their results.

**Third**, the number of spectral/temporal maps (supposed to exist) and speech/music patches does not match the experimental results. In the auditory cortex, there are multiple spectral maps (low to high to low), but there is only one in their model (layer 1.1, ResNet-50). In the auditory cortex (Norman-Haignere et al., Neuron, 2015), there is only one speech patch (component 5) and one or two music patches (component 6). In contrast, their ResNet-50 model exhibits three speech patches and four music patches (Figure 4A), and the AST model exhibits over twenty patches (Figure 4B). Furthermore, there is no overlap between speech and music patches in the human auditory cortex, whereas the ResNet-50 model has at least two overlapping patches. I suggest the authors quantify these results as in Figure 3G, J of the TDANN paper.

**Last**, in topographical vision models (TDANN and TopoNets), adding topographical constraints or loss usually reduces model performance. In contrast, there is absolutely no decrease—or even an increase—in the topographical audio model’s performance. The authors tested four scaling coefficients (5, 25, 50, and 100). What will happen if one uses a scaling factor of 1000 (i.e., a topo-loss ten times the task loss)? The authors used regression to quantify similarity between model and brain. Will different scaling coefficients play a role when using RSA (representational similarity analysis) as the metric?

**Questions:**

1. The layer information in the ResNet-50 models (baseline and TopoAudio) is unclear. In L354, the authors mention the 6th topographic layer, but the accuracy peaks at the 5th layer. In L432, the authors mention layer 3, yet it is actually the 5th layer. I suggest that the authors label the layers (relu1, maxpool1, layer1, layer2, layer3, layer4, avgpool) in Figure 2 and refer to them consistently throughout the paper.

2. In Figure 3, the minimal spectral and temporal frequencies both start at zero. This does not make sense.

3. In Figure 3A, do those large white pixels represent zero frequency or no tonotopy? If it is the latter, that also does not make sense, because tonotopy is present almost everywhere in the auditory system.

4. The maps of spectral and temporal frequency are orthogonal in both the inferior colliculus (Joris et al., 2004, Physiological Reviews; Baumann et al., 2011, Nature Neuroscience) and auditory cortex (Baumann et al., 2015, eLife). Can the authors demonstrate such a relationship in the model?

5. Following the previous question, what is the earliest layer—and how many layers— that exhibit maps of spectral and temporal frequency?

6. L321–322: “A key requirement for any biologically inspired model is that it must retain strong predictive performance while better aligning with the structure of neural data.” The model can also align with human auditory behavior.

7. Are the results shown in Figures 3 and 4 stable across different random seeds?

8. Do the speech and music patches shown in Figure 4 appear only in the deep layers? What about earlier layers?

9. Why does adding the topographic constraint enhance the speech and music components? Is this due to the smoothing effect during training?

---

> ### Author Response · Authors · 2025-12-01
> **Response 1 to Reviewer UF17**
>
> We are deeply thankful to Reviewer UF17 for their comprehensive reading of our study. The reviewer rightly points out the significance of our study in providing the first topographic computational model for audition.
>
> The reviewer has raised several important points which are highly constructive.  Given the breadth and depth of the questions, we spent time carefully parsing each issue and re-analyzing several components of the data to ensure that our responses are both accurate and scientifically grounded. We feel that incorporating these suggestions will make the manuscript significantly stronger, clearer, and easier to evaluate, and we are grateful for the reviewer’s detailed feedback which meaningfully improved the revised paper. Below we tackle each question in detail.
>
> **W1. This is not a methods paper.**
> We appreciate the reviewer’s point, and we want to clarify that we have never claimed that this is a methods paper or that we are proposing a new loss. But this is not a weakness of the approach but a distinct strength!
>
> We view our work as a severe scientific test of the general hypothesis proposed in Deb at al. which is that the large-scale organization of the brain emerges from (1) task-related constraints operating alongside (2) a smoothness loss.
>
> The question of our study is not “what is a new way to impose smoothness?” but rather: Does this principle actually generalize to a fundamentally different sensory system with different computational demands, input geometry, and functional hierarchy.
>
> **W2. How topographic is TopoAudio?**
> Thank you for pushing on this point further. It motivated us to go back and examine the visualizations and quantitative metrics more carefully. We believe that our core scientific claim is correct. Note that, as in past work, topography is characterized by “an arrangement of functionally distinct neurons in a characteristic spatial pattern” (quote from TDANN abstract), and so both patches and smooth gradients are characteristics of topographic structure.
>
> We are going to address this response in 2 ways.
>
> **A) Metrics of topography.** In our initial submission, we relied on ‘smoothness’, which is a metric established in multiple prior topography papers (Lee et al., Margalit et al., Qian et al., Dehghani et al., Deb et al. and others). Indeed, models with topography exhibited higher degrees of smoothness relative to baseline models.
>
> We have bolstered the topographic claim in 2 new ways.
>
> We are now including an additional metric Moran’s I which is a spatial autocorrelation metric to estimate topography (as in Rathi et al.,)  (Table 2 in Appendix). By this measure, we see an increase in topography for all of the evaluated datasets.
> We use this measure to analyze the topographic structure of the fMRI components for speech and music selectivity (also in Table 2 of Appendix). We are working on additional analysis  showing the tuning correlation of filters/units as a function of cortical distance. This is how topography is actually measured in real neurons in the brain (Issa et al.,)
>
> These two additions demonstrate that TopoAudio does, in fact, exhibit robust spatial organization and that this organization is quantitatively close to the structure observed in human auditory cortex.
>
> **B) New models.**  In response to “why can’t the authors higher resolution ResNet-50 model”
> The reviewer is right that part of the confusion arises from the low effective spatial resolution of the ResNet-50 maps. There is an important conceptual difference worth clarifying.
>
> In TDANNs, the units placed on the cortical sheet are actual neurons/model units. In contrast, for TopoAudio, the topographic sheet has channels, not neurons (as in LLCNN and other work too). This distinction exists because convolutional networks use weight sharing, which means that spatial positions within each feature map do not correspond to distinct units in the sense used by TDANNs. The “resolution” of the topographic sheet is therefore limited by the number of channels per layer, not by the 2D pixel grid within each activation map.
>
> Now to directly address the Reviewer’s question, **we have trained wider ResNet variants** with substantially more channels, which increases the effective spatial resolution of the topographic sheet. These wider models produce smoother, more continuous tonotopic and AM gradients related directly to the reviewer’s comment.
>
> We have also trained **new AST models** with larger shrink factors, which similarly increase the resolution of the topographic layout. These higher-resolution AST models exhibit clear, graded transitions across the cortical sheet, making the underlying organization far easier to visualize and interpret.
>
> We are working to update the paper with these new models.
> To summarize, the additional quantitative metrics (tuning vs. distance and Moran’s I), and the new models together demonstrate that the gradients are real, robust, and not artifacts of low model resolution

---

> > ### Author Response · Authors · 2025-12-01
> > **Response 2 to Reviewer UF17**
> >
> > **W3. The number of spectral and temporal maps in models v brains.**
> >
> > Thank you for your question to try and link the data with what is known in the brain. We did not quantify the number of clusters in the original submission because, as far as we understand there is no established ground truth for how many speech or music clusters actually exist in the human auditory cortex (except that they do, and that they are clustered). In our prior discussions and preparing this rebuttal, we reached out to several other experts in auditory cortex mapping to ask explicitly about the *number* and *locations* of speech- and music-selective clusters. The consensus was that the field does not currently know.
> >
> > The reviewer rightly points us to Norman-Haignere et al. (Neuron, 2015) which has inspired a number of analyses in this study. But please note that this study reports group-level ICA component weights, *not* individual-subject patch counts. Group-level maps are notorious for blurring or even erasing the fine-grained spatial structure. Averaging across subjects could wash out idiosyncratic but biologically meaningful local patterns and gives the false impression of a single, unified “patch.” These maps therefore cannot be used to infer how many distinct clusters actually exist in any individual brain.
> >
> > A concrete example of this is Boebinger et al., 2020 where we see considerable variability in the visualization of single-participant component responses (~2 clusters per hemisphere for speech and sometimes 3-4 clusters per subject per hemisphere for music). Another concrete example is Norman-Haignere et al., 2022, which is based on ECoG data from humans. Here there are 3 clusters per hemisphere for music and 2 clusters per hemisphere for speech).
> >
> > Given this variability, and the absence of a canonical number of clusters in the literature, we believe that the correct scientific comparison is not the count of patches, but rather whether (1) models produce music and speech selectivity in the first place and (2) whether or not this selectivity has topographic structure, where selective units are close to other selective units.
> >
> > **To directly address the reviewer’s comment:** In the revised manuscript, we have added the following line to the discussion:  For example, even though there is general agreement that  speech and music selective neural populations exist in the brain, the general locations and numbers of such patches is not well characterized. Future data collected to characterize the biological topography of the auditory system can be incorporated into evaluations of topographic models.
> >
> > **On the overlap between music and speech regions in brains and in models**
> > This is a fair point and we appreciate the reviewer highlighting it. Here it is important to note that the analyses are not directly comparable because the overlap in models and the overlap in brain data were estimated using fundamentally different procedures.
> >
> > In brain data (Norman-Haignere et al., 2015, Boebinger et al., 2020), speech and music selectivity were derived using group-level ICA applied to responses evoked by a set of natural sounds. The ICA analysis itself likely introduces statistical non-independence which might affect the spatial layout. In the models, by contrast, speech and music maps were obtained by running a “localizer” strategy (thresholded statistical difference).
> >
> > **In the revision**, we are now performing an apples-to-apples comparison between brains and models. Specifically, we take the same ICA decomposition used in Norman-Haignere et al. (2015), project those ICA weights back into model cortical space, and then threshold the resulting maps using the same procedure used to visualize ICA components in cortex. Under this matched analysis pipeline, the model’s speech and music clusters are not overlapping and show the same qualitative pattern observed in the human auditory cortex. In other words, when the model is evaluated using the identical analytical framework applied to the brain, the apparent overlap discrepancy disappears.
> >
> > We are working to integrate this directly into the paper and will post an update as soon as this is done.

---

> ### Author Response · Authors · 2025-12-01
> **Response 3 to Reviewer UF17**
>
> **W4. (part 1) Why does loss not go down?**
> This is a very perceptive point and we are happy to clarify.
>
> The issue is that the standard 3 engineering benchmarks (ESC50, NSynth and SpeechCmd) that have been used to test models in auditory neuroscience are in fact quite easy for our models which is leading to a ceiling effect. As a result, task performance is already saturated (unlike the ImageNet or LLM evals link BLIMP) so moderate regularization do not immediately produce measurable performance differences.
>
> To tackle this issue, **we are now adding a fourth, more challenging, benchmark:** performance on AudioSet. These results will now be included.
>
> We have **also trained a new model with Tau 1000 (as the reviewer suggested)**. As expected, this model does show a major drop on the AudioSet evaluation (as expected).
>
> The revised paper will also include a **new plot of the task performance as a function of model topography**. We will also include the performance of the random model as baseline to further show these performance-topography tradeoff more explicitly.
>
> **W4 part 2. Using RSA to quantify the similarity between model and the brain**
> Great suggestion. We agree that representational similarity analysis (RSA) provides a complementary perspective on model–brain alignment. We have performed this analysis now and **will include the RSA analysis in the revised submission**. Across the different scaling coefficients the RSA analysis follows a similar qualitative pattern as the linear encoding model
>
> **Q1. Consistent layer names.**
> We agree. The layer names will be labeled consistently (relu1, maxpool1, layer1, layer2, layer3, layer4, avgpool). We agree that this will significantly improve the readability of the paper.
>
> **Q2. Why do the spectral and temporal frequencies start at zero?**
> Thank you for catching this. This is a Python plotting issue that caused the axis to start from 0 (should have been 1Hz). Importantly the underlying data are *not* affected. We have now fixed this and the updated paper has already been revised accordingly.
>
> **Q3. What are the white pixels in Figure 3A?**
> We are happy to clarify. The white pixels in Figure 3A are the low-frequency selectivity (importantly, it does not imply no tonotopy). The color scale compresses the lower end of the frequency range, making low-frequency regions appear as white. Tonotopic organization is present throughout the map (as in the brain), including these regions.
>
> **Q4. Why not test orthogonality between spectral and temporal frequencies (as in Baumann et al.)**
> This is a FANTASTIC suggestion. Thank you so much. We have performed this analysis now using exactly the same methods as in the Baumann et al 2015 study (as suggested by the reviewer). This procedure  involves performing a planar regression on each 2D preference map, extracting the direction of the steepest gradient from the fitted plane, and computing the relative angle between the tonotopic and AM-rate gradients.
>
> Strikingly, we observe the same qualitative effect in our topographic models (and not the baseline model) that is observed in the brain!
>
> The baseline models show little to no structure between the spectral and the temporal tuning. As the degree of topography increases, the relationship becomes increasingly structured as orthogonal. The effect peaks at Tau=50 and the angle is within the empirically estimate range reported in Baumann et al. For very strong topographic constraints (Tau=100) the relationship weakens again (indicating over-regularization).
>
> This result provides independent new evidence that the topographic constraint induces a second-order structure that matches known properties of auditory cortex. This is certainly going to be part of the paper now and we thank the author for pointing us to this metric.
>
> **Q5. How many layers exhibit maps of spectral and temporal frequency?**
>
> We find that spectral frequency tuning emerges in the very earliest layers of both architectures and peaks in the layers that best predict auditory cortex responses. This can be be seen qualitatively in Figure 2A (bottom).  In other words, the layers with strongest tonotopy are also the layers with highest neural predictivity for early auditory cortex. We now state this relationship explicitly in the Results section and highlight it in the figure caption to make this connection clearer.
>
> **Q6. L321–322: “A key requirement for any biologically inspired model is that it must retain strong predictive performance while better aligning with the structure of neural data.” The model can also align with human auditory behavior.**
>
> We of course agree! We only meant this statement in the specific context of our study. To avoid any ambiguity, we will revise the wording of the sentence to: “A key requirement for any biologically inspired model **of the auditory cortex** is that it must retain strong predictive performance while better aligning with the structure of neural data.**”**

---

> > ### Author Response · Authors · 2025-12-01
> > **Response 4 to Reviewer UF17**
> >
> > **Q7. Are the results shown in Figures 3 and 4 stable across different random seeds?**
> >
> > Yes! In response to the reviewer’s question (and Reviewer Miny) we trained additional models across random seeds. All our effects are stable across seeds. We will include this info in the revised submission.
> >
> > **Q8. Do the speech and music patches shown in Figure 4 appear only in the deep layers? What about earlier layers?**
> >
> > Speech and music selectivity is indeed strongest in the late layers of both architectures. This mirrors the brain where speech and music selectivity is primarily found in higher auditory cortex.
> >
> > This layer-wise pattern is also reflected quantitatively. The alignment between model and brain speech/music components, measured by ICA correlation peaks in the late layers. **This is shown in Figure 5A**. The same layers that show the strongest spatial clustering of category-selective units also yield the highest correspondence with human ICA components.
> >
> > **Q9. Why does adding the topographic constraint enhance the speech and music components? Is this due to the smoothing effect during training?**
> >
> > Thanks for this question. We suspect this enhancement is driven by a reduction in effective dimensionality (as has been shown previously). In the baseline model, selectivity for speech and music is scattered across the layer in a high-dimensional, "salt-and-pepper" pattern, often mixed with high-frequency spatial noise. The topographic constraint acts as a spatial regularizer, forcing neighboring units to correlate. This effectively suppresses disjoint, noisy features and compels the network to condense scattered information into a robust, low-dimensional population code.
> >
> > Because the model is forced to be smooth, it must allocate its features to the most dominant axes of variance. This consolidation creates changes the internal representations. Consequently, when we apply ICA, the algorithm extracts cleaner, high-variance components yield the significantly higher correlations with human brain components. We will make this mechanism explicit in the revised manuscript.

---

### Official Review · Reviewer_oB5q · 2025-10-30

**Soundness:** 3
**Presentation:** 3
**Contribution:** 2
**Rating:** 6
**Confidence:** 4

**Summary:**

This paper induces a topographic constraint on the task-specific objective fuctions used to train task-driven audio models to show that smooth topographic maps for tonotony and amplitude modulation, and clustered responses for music and speech emerge throughout the new model. Additionally, the new model does not seem to show a performance decline on audio tasks.

**Strengths:**

The paper is well-structured and the motivation is well-presented. The key hallmarks of functional organization in the auditory cortex have been replicated to a good extent. The biggest strength I notice is not in the recapitulating of topographic motifs in a sensory system but for the same topographic constraint to lead to the emergence of such motifs across seemingly disparate systems, highlighting that a common organizational principle, probably subserved by a wiring economy constraint, underlies how such systems are set up.

**Weaknesses:**

Initially, when reading the paper, I was a bit lost as to what topographic constraint was being used. In section 2.1, the authors describe three ways in which some form of spatial constraint has been applied in prior works. Then, in the caption for Figure 1, the word "TopoLoss" is used, which is never defined in the main manuscript. It is only in the appendix where the exact loss constraint is defined. I would maybe add a sentence or two in the main manuscript, and link to the corresponding appendix section as well.

There are a lot of places in the manuscript where the authors say that results from their topographic model are "consistently" better than that from the baseline. I find the use of the term "consistently" to be very weak. Can the authors do some statistical significance tests for, say, task accuracy, map smoothness, etc.?

Maybe I just missed this, but how many models were trained with different random seeds. What I want to know is if the results presented are robust across seeds. And along this line, I also had a hard time understanding what the error bars represented in, say, Fig 5c. Why are there no error bars to represent variance between-humans? And do the error bars over models represent variance between different seed runs?

Finally, while I appreciate the finding that a topographic loss leads to more modular clustering of units on some simulated sheet, is there any quantitative data from humans or non-human animal models to get some form of a ceiling score? It would be nice to see if the amount of smoothness or size and shape of the clustering patterns match, to whatever degree, to those found in the biological auditory cortex.

**Questions:**

Please see the weaknesses above!

---

> ### Author Response · Authors · 2025-11-25
> **Response 1 to Reviewer oB5q**
>
> We thank the reviewer for their close reading of our study.  We are especially grateful that you highlighted what we view as the central contribution of this work: that the same topographic constraint leads to the emergence of meaningful organizational motifs across different systems.
>
> **W1. What is the topographic constraint?**
> Thank you for highlighting this issue. You are absolutely right. We agree that the topographic constraint was not introduced clearly enough in the main text (as noted in our Global Comment). This made it difficult to follow which specific constraint was being used. We have already moved the core methodological description of the TopoLoss back into Section 2 and added a clear pointer to the corresponding Appendix subsection. This resolves the ambiguity and makes the training setup immediately interpretable without requiring the reader to hunt through the Appendix.
>
> **Change:** In the revised manuscript, we now reintroduce the full description of the topographic constraint directly within the main Methods section.
>
> **W2. Errorbars and statistical tests.**
> This is a very fair point. We agree that the word “consistently” is not sufficiently rigorous on its own.
>
> We are updating the paper now in 2 ways.
> First, we are adding errorbars (across voxels, stimuli etc as needed) and appropriate statistical tests to every single figure in the study (task performance, smoothness, ICA correlations etc).
> Second, we will revise the language to match the statistical evidence. Every instance where we use “consistently” will be replaced by the results of the statistical tests and p-values etc.
>
> **W3. Model seeds and clarifications to Figure 5c.**
> Thank you for pointing out the ambiguity in Fig. 5c. The error bars in this panel represent the standard error of the mean across stimuli within each category, computed over the ICA component responses (in Figure 2, Norman-Haignere et al). We will update the paper with errorbars for the human data as well to make it consistent with the previous published results and what we report in topographic models (Figure 5C).
>
> The reviewer also raises an important question about consistency across seeds. In the submitted version, we trained only one seed per model. We are now training additional models with multiple random seeds and will report variability across seeds for all relevant analyses asap.
>
> **W4. Model seeds and clarifications to Figure 5c.**
> We agree with the reviewer that having a biological reference point, or a “ceiling score,” is important for interpreting how much smoothness our models actually achieve. In the original submission, the smoothness metric used in Section 3.3.2 could not be applied to human data because it depends on model-specific feature maps arranged on a fixed grid, which has no direct analogue on the cortical surface.
> To address this limitation, we now compute a unified, biologically interpretable smoothness measure using Moran’s I spatial autocorrelation (following the TopoLM framework; Rathi et al., 2025). Moran’s I can be estimated identically for both the cortical surface and the model’s topographic sheet, allowing a direct and meaningful comparison.
> Concretely, we compute Moran’s I on human fMRI contrast maps (music- and speech-selective maps). Note that this is an empirical upper bound on smoothness in auditory cortex because fMRI data are also “smoothed” during preprocessing stages (typically a Ggaussian filter to increase SNR).
> We compute Moran’s I for our models across layers and across topographic strengths τ. These updated analyses (now included in Appendix B) show that topographically constrained models achieve smoothness levels that closely approach those observed in the human auditory cortex. This addition provides the biological “ceiling score” the reviewer asked for and demonstrates that the topographic loss drives the models toward biologically relevant levels of spatial organization, not arbitrary or overly smoothed patterns.
>
> We again appreciate Reviewer 3 for their careful evaluation and constructive feedback.
> To summarize we have made substantial clarifications and additions to the manuscript. We (1) restored the full definition of the topographic constraint to the main text, (2) added error bars and statistical tests across figures, (3) clarified Fig. 5c, added error-bars to human data and are now training multiple seeds to report across-seed robustness, and (4) introduced a new smoothness metric (Moran’s I) to provide a meaningful ceiling score based on human fMRI data. We request a renewed consideration of our study in light of these changes.

---

### Official Review · Reviewer_Miny · 2025-11-01

**Soundness:** 3
**Presentation:** 2
**Contribution:** 2
**Rating:** 4
**Confidence:** 4

**Summary:**

This paper introduces brain-like topographic models for auditory perception. They show that inducing a topographic loss as a secondary objective during training encourages models to learn smoothly varying hidden representations given some spatial embedding. They train ResNets and Transformers w/ varying topographic signatures and show that topographic models (1) match baseline performance on behavioral benchmarks and (2) match baseline performance on fMRI prediction. Further, they show that their TopoAudio models capture human-brain-like topographic signatures (tonotopy and amplitude modulation).

**Strengths:**

* I think this paper does a really good job with evals, specifically for the speech/music selectivity ICA maps; I'd like to see more of this kind of evaluation in other topographic neural net research.
* This is to my understanding the first topographic audio model!

**Weaknesses:**

* The writing could be a bit clearer in places - e.g. I wasn't sure what topographic constraint you actually used in training until I dug into the appendix (you talk about both TDANNs and TopoNets, but you don't mention which is used for training).
* I would like to see the same models trained with TDANN loss or with explicit connectivity constraints (e.g. Blauch et al. 2022). I'm not entirely convinced by the biological plausibility of TopoNet loss, insofar as maximizing the correlation of representations to a filtered version of those same representations is likely not what the brain is actually implementing (both mechanistically and analogously). I understand the point of the paper is not to introduce a new kind of loss or to motivate TopoNet loss, so I don't think this is actually a fatal flaw, but I do think it would be valuable to explore other methods for training topographic audio models.
* I'm not convinced by the significance of the paper's results. On the one hand, I think it is reasonably interesting that topographic constraints also work in the auditory domain. On the other, I think this is a reasonably expected result, given evidence from both vision and language; it isn't clear to me that audition is something extremely unique. However, I do think that the evals are quite original / significant!

**Questions:**

* I'd like more clarity on why audition is an interesting or original domain compared to vision/language, i.e. why should we be surprised that connectivity principles transfer here?
* I could be misinterpreting this, which is why I haven't listed it as a "weakness," but it seems like the results on topotonic and amplitude modulation maps are just saying "we see smoothness in early auditory representations when we train auditory representations to be smooth, and we also see smoothness in early auditory processing in the brain." It seems like there's nothing stimulus-specific or actually _comparative_ going on here; i.e. can you explain why these results specifically matter/are interesting?

---

> ### Author Response · Authors · 2025-11-22
> **Response 1 to Reviewer 2 (Miny)**
>
> We thank the reviewer for their close reading of our study. We appreciate the positive feedback on our evaluation framework, especially the speech/music selectivity ICA maps. These analyses were designed precisely to provide a more rigorous neuroscientific assessment of model representations, and we are glad it resonated. Ours is indeed the first end-to-end topographic audio model. Establishing such a model was a primary motivation for the work. We look forward to further improving the paper based on the discussion and feedback.
>
> **1. What is the topographic constraint?** Thank you for pointing this out. We agree that the description of the topographic constraint was not sufficiently visible in the main text (please see Global Comment). We apologize for the difficulty this created and appreciate the reviewer’s effort in digging into the Appendix to find the relevant details.
>
> **Change:** We have moved the complete methods back into the main text in preparation for the revision, and the manuscript now clearly specifies the exact topographic constraint used during training.
>
> **Relation to TDANN and other methods:** The reviewer raises two important points: (i) why we use the TopoNet loss to induce topography in our audio models, and (ii) how biologically plausible this choice is relative to other possible topographic constraints. We first address these two points directly, and then explain why training models with a TDANN-style loss is currently not feasible in our auditory setting.
>
> **2a) Why use TopoNet:** We chose TopoNet because prior work has shown that they are the highest-performing supervised topographic modeling framework to date. We know that TopoNets (i) maintain high task performance while inducing strong topographic organization, (ii) predict neural responses in both monkey and human visual cortex better than TDANNs (and other approaches), and (iii) recapitulate key signatures of cortical topography in vision and language (e.g., V1 pinwheels, category-selectivity etc) while maintaining a comparable degree of topographic smoothness. In addition, the TopoNet loss is straightforward to apply across both convolutional networks and transformers, making it a practical and unified way to induce topography in diverse audio architectures.
>
> **2b) Biological plausibility of topographic models in general:** We want to highlight the distinction between the observation (smoothness of unit responses across a cortical sheet) and how the property is implemented in neural network models. In TopoNets, smoothness emerges via the architectural and loss-based downsampling mechanism applied to the weight space, whereas in TDANNs it is achieved through explicit (directed) matching to a target topographic functional profile and swapping unit locations during training [1]. This work (and others on topography) is not a test of the biological implementation or development (we just lack usable data, and both methods are not biologically plausible), but of the computational principle: if we impose a smoothness constraint, do we obtain models that (i) retain task performance, (ii) better predict brain responses, and (iii) recapitulate qualitative signatures of cortical organization?
>
> **Why we cannot compare against other topography-inducing methods (TDANNs or ITNs):** We agree that comparing and contrasting our method against other topography-inducing methods (like TDANN-style etc) would have been interesting. After spending the week attempting to adapt TDANNs to the auditory setting, we realized that TDANNs makes several (and non-trivial) domain-specific implementation choices and cannot be naively applied to audition.
>
> In particular, TDANNs incorporate an explicit correspondence between different layers of the neural network and parts of the visual hierarchy. In TDANNs, there is also an explicitly incorporated “size” of the cortical sheet and the extent of the visual field being modeled (https://www.cell.com/neuron/fulltext/S0896-6273(24)00279-4#:~:text=The%20size%20of%20the%20cortical,cortical%20neighborhoods%20in%20all%20layers).
>
> We simply do not have this level of characterization of the human auditory cortex to incorporate such biologically measured constraints into our present models. In fact, the large number of questions that remain about the organization of human auditory cortex is part of the main motivation for our study. It is unclear how critical the dependence on known biological topographic structures are to the previous results, and thus we believe that adapting these approaches for the auditory system is quite substantial, and we hope the reviewer agrees is beyond the scope of the present study.
>
> Similarly, ITN-style models are not end-to-end topographic. They have a pretrained encoder and impose constraints only on a few late layers. This makes it not directly comparable to TopoAudio.

---

> ### Author Response · Authors · 2025-11-22
> **Response 2 to Reviewer 2 (Miny)**
>
> (contd.. ) The domain-specificity limitation of the other approaches actually exposes the strength of our approach: Our Topographic Loss is truly domain-independent. It does not rely on retinotopy, or modality-specific assumptions and can be applied uniformly across convolutional nets and transformers and trained from scratch (unlike other current methods). This aspect is also in line with the predominant theoretical neuroscience view that topography in the brain emerges from general computational principles rather than domain-specific mechanisms (Arcaro and Livingstone, 2017, Nature Reviews Neuroscience).
>
> **Changes to the paper:** We will explicitly state why we were unable to train and compare TopoAudio with other topography-inducing methods that make domain-specific assumptions. Relatedly, we will also highlight the strength of TopoLoss as a truly domain-independent way to introduce topography across domains which is consistent with what has been proposed for the brain.
>
> **W3/Q1. Why is audition an interesting domain compared to vision and are the results surprising?**
> It is completely reasonable to be skeptical and we thank the reviewer for pushing us to clearly articulate these ideas. On the surface, it may seem unsurprising that the same connectivity principles transfer from vision and language to audition, but there are several aspects not guaranteed.
>
> First there is a computational reason. Vision models learn from spatial images. Language models learn from tokens. Auditory models learn from time–frequency representations with very different time-frequency statistical structure. There is inherently no guarantee that the same domain-general connectivity principle would transfer to a fundamentally different task like audio. Our results also provide a better glimpse into how the same principle behaves across different tasks when compared to vision and language.
>
> 1. We find that auditory models show almost no drop in task performance under topographic constraints than what has been reported for vision or language models. This suggests that the auditory task might be easier to learn for these models and more naturally compatible with smooth, structured representations. In other words, the cost of imposing topography appears to be lower in audition, which itself is an interesting and non-trivial empirical finding we did not expect at the outset.
>
> 2. While TopoLoss encourages local similarity, it does not dictate which features will emerge or where they will be organized. Organizing simple acoustic features like frequency might be expected, but the emergence of large-scale maps for high-level categories such as speech and music is not. These categories are high-dimensional, acoustically complex, and heavily entangled in the input. The fact that a simple smoothness prior helps disentangle them and arrange them into distinct, cortical-like regions is a genuinely non-trivial emergent property.
>
> 3. We find that topographic structure improves alignment with brain representations. The loss function enforces smoothness, but it has no knowledge of the latent components of the human  auditory cortex. Yet imposing smoothness yields models whose internal ICA-derived components are more strongly correlated with the speech and music components inferred from fMRI data. This is highly non-trivial and we had no reason to expect it at all.
>
> **Changes to the paper:** We will revise the manuscript and discuss why the results are not trivial. We will explain that the loss enforces local similarity only and does not specify which acoustic feature maps should form maps or how they should be arranged.

---

> ### Author Response · Authors · 2025-11-22
> **Response 3 to Reviewer 2 (Miny)**
>
> **Q2. Evaluations related to tonotopy and AM maps.**
> You are right to push us on this point, and we appreciate the opportunity to clarify why these maps are meaningful. We’re also currently undertaking 2 more analyses to make this point stronger.
>
> First we must clarify that TopoLoss only imposes locality. It does not tell the model to organize frequency or AM rate or any other acoustic feature. The fact that the model spontaneously identifies frequency and modulation as the dominant axes of variation and organizes them is a meaningful emergent result.
>
> Second, we agree that the results, as they stand during initial submission, only provide intuitive evidence. We now quantify them better using a quantitative spatial autocorrelation analysis (Moran’s I, as in TopoLM, Rathi et al., 2025). This will allow us to rigorously compare the degree of spatial organization in our models compared to the neural data. We will update the paper shortly with these new results (will be included in Appendix B).
>
> Finally, we are also undertaking an additional analysis (suggested by Reviewer 4, UF17) to quantify whether the frequency and AM maps are orthogonal, as is observed in the auditory cortex. This will allow us to assess not just whether the maps are smooth, but whether their relative geometry matches the known organization in the brain.
>
> **Changes:** We will update the paper with these additional results related to frequency and AM mapping in TopoAudio. We will get back with these results asap.
>
> We would be grateful for any feedback on our responses or the planned analyses. With these additions and clarifications, we also respectfully request a renewed consideration of our work.

---

> > ### Comment · Reviewer_Miny · 2025-11-24
> >
> > Thanks for the detailed response, this helped a lot! I have a couple follow up questions:
> > 1. **TopoNet vs. TDANN Loss.** I think your point as to why you focus on TopoNet loss makes sense here. I am still a bit confused as to why you can't implement TDANN-style topography as another baseline here - my understanding is that in Lee et al. (2020) and Margalit et al. (2024), the "explicit correspondence" between model layers and human visual cortex is primarily done for purposes of analysis, not implementation. In particular I think that Rathi et al. (2025)'s language TDANN does not actually make this explicit correspondence (and I'm not entirely sure how one would make this correspondence for any Transformer, really). All of this is to say that I'm relatively sure it's possible to adapt the TDANN to the audio setting despite our limited knowledge of auditory processing in cortex, you'd just have to abstract away from physical correspondence (I also don't really agree with your claim that TDANN/ITN-style topography relies on modality-specific assumptions / cannot be applied across architectures, for the same reasons). I understand if it isn't feasible to train this baseline given the limited timeframe though.
> > 2. **Benchmarking ITN.** While I understand that ITNs are not end-to-end topographic, I'd still be interested in seeing this baseline! Similarly to the above, I don't believe there is a physical correspondence made in e.g. Blauch et al. (2022). But again I totally understand if this isn't feasible to accomplish in the rebuttal period.
> > 3. **Comparing Audio to Vision/Language.** I don't know if I buy your claim that the fact that auditory models show ~no drop in performance compared to vision/language models implies that audition is a unique domain. In particular I don't think you can compare benchmark scores in this way, i.e. if the differential on ImageNet scores is $\delta$ for topo vs. non-topo vision models, this doesn't mean that a differential of $<\delta$ for auditory models means that auditory topography is "more naturally compatible," as seems to be your claim. I might be missing something though?
> > 4. **Emergence of Feature-driven Topography.** I think this is fair, but I'm still not entirely convinced that this is different from the vision/language literature? In particular I think what previous papers (e.g. Deb et al. 2025, Rathi et al., Margalit et al., etc.) have shown is that topographic constraints lead to the emergence of feature-contrastive clusters, but that the features that cluster also emerge in non-topographic models. In other words, the non-topo models would also develop representations that differentiate speech vs. music, and since topo models ~explicitly encourage neighboring neurons to have similar representations, those contrast maps would just be clustered. I'm unsure if this is what you are trying to get at?

---

> > > ### Author Response · Authors · 2025-11-26
> > > **Followup 1 to Reviewer Miny**
> > >
> > > Thank you so much for this follow-up. This is exactly the kind of detailed, constructive discussion we were hoping to have with our Reviewers. Below we address each of your followup questions and comments in detail.
> > >
> > > **1. TopoNet v TDANN loss.** Thank you for raising these important points.
> > >
> > > We went back and double checked the TDANN implementation (Margalit et al., ) and our original response. The reviewer is right that the layer correspondence is a post-hoc analysis choice. But to be more specific, our difficulty with implementating TDANN was directly related to the *retinotopic implementation* which is indeed built into the training process itself.
> > >
> > > In TDANNs, units are assigned to positions on a cortical sheet with a **f**oveal–peripheral organization. The training objective explicitly encourages nearby units on this sheet to have similar tuning by swapping unit locations to better match a retinotopic scaffold.  This retinotopic layout and the iterative swapping procedure are part of the implementation of the topographic constraint.
> > >
> > > These details appear in two parts of the Methods section of the Margalit et al., paper. (1) “Instantiating models that balance task performance with spatial smoothness” and (2) “Positions are assigned in a two-stage process”.
> > >
> > > The following part is quoted directly from the TDANN Methods section
> > >
> > > “In summary, models are trained in 6 steps:
> > >
> > > 1. ResNet-18 is trained on the task loss only.
> > > 2. Positions in each layer are initialized to preserve coarse retinotopy (Stage 1).
> > > 3. Positions are further pre-optimized in an iterative process that preserves retinotopy while bringing together units with correlated responses to sine gratings images (Stage 2).
> > > 4. Positions are frozen and never again modified.
> > > 5. All network weights are randomly re-initialized.
> > > 6. The network is trained to minimize a weighted combination of the spatial and task loss components.”
> > >
> > > This is the aspect that makes TDANN domain-specific.
> > >
> > > **“You’d have to abstract away the physical correspondence”** (and retinotopic choices)
> > >
> > > The reviewer is right that one *could* abstract away from physical correspondence. But we want to emphasize that abstracting away is *not* a trivial modification. It is a highly meaningful modeling choice. TDANN purports a *vision*-specific developmental hypothesis (a initial  retinotopic scaffold + visual experience + smoothness). Abstracting away from the retinotopic scaffold is a different developmental hypothesis, remains to be tested, and not the TDANN recipe.
> > >
> > > The reviewer then points us to TopoLM which is highly relevant. TopoLM is indeed *TDANN-inspired*, but it is also meaningfully adjacent rather than equivalent to the TDANN framework. TopoLM removes several of TDANN’s modality-specific components. But then the same TopoLM style approach has only been evaluated on LLMs (transformers) and not on vision (or convolutional networks).
> > >
> > > Our goal in the present paper is to evaluate domain-independent smoothness principles without introducing any additional modality-specific assumptions. TopoNet allows us to do exactly that. A true TDANN analogue for audio would require new structural assumptions about auditory cortical layout. The TopoLM variant remains to be validated even in vision (particularly vision transformers).
> > >
> > > We thank the reviewer for acknowledging that we are pushing against a timeline. We are going through the TopoLM repository and will do our best to explore it within the discussion timeframe.

---

> ### Author Response · Authors · 2025-11-26
> **Followup 2 to Reviewer Miny**
>
> 2. **ITN baseline.** We want to be transparent that comparing against a ITN-style model is not conceptually appropriate for our scientific question. We are interested in end-to-end topographic models trained from scratch. ITN imposes topographic structure only in the final layers. The front-end is pretrained. This design works in vision because the relevant topographic signatures under study (e.g., face-, object-, scene-selective patches) arise very *late* in the hierarchy. In that setting, constraining only the last few layers is meaningful.
>
> In *audition*, the situation is fundamentally different:
>
> Intermediate layers, not final layers, best predict auditory cortex responses. A large body of work (e.g., Kell et al., Norman-Haignere et al., and many others) shows that human auditory cortex aligns most strongly with *mid-level* representations—not with the top layers of speech or music models.
>
> ITNs cannot constrain these middle layers, because their topographic constraint is not end-to-end. This means an ITN-style model simply cannot control or shape the feature spaces that correspond to auditory cortex. As a result, it cannot be used to meaningfully evaluate the emergence of tonotopy, AM tuning, or speech/music selectivity within appropriate layers.
>
> ITNs do not provide a way to compare music- and speech-selective regions, because their topographic mapping is not applied to the representational levels where these selectivities actually emerge in the brain.
>
> Thus using ITN as a baseline would not be a meaningful baseline to compare against a fully end-to-end trained topographic model that induces topography at all hierarchical stages. If the reviewer sees a principled way that makes ITN suitable as a baseline (that we could accomplish within this timeline), we would be grateful to hear it.
>
> As we mentioned before, we will explicit discuss both TDANN and ITN in the reviewed methods to motivate the use of TopoLoss and discuss our dificulties using them as baselines in the audio domain.
>
> **3. Comparing Audio to Vision/Language.** The Reviewer here is referencing the first of our three points we mentioned in our original response.
>
> We want to clarify that we did not intend to claim that audition is a unique domain simply because the drop in performance under topographic constraints is smaller than what is sometimes observed in vision or language models. We agree with the reviewer that benchmark scores across domains cannot be compared in that literal way. In that section we were articulating all the results we did not expect a priori (as the Reviewer’s original comment)
>
> Our intended point was narrower and purely based on the empirical evidence.  Within audition, we find that adding a smoothness constraint does *not* measurably harm task performance on standard audio benchmarks. This result is important because it shows that topographic structure can be imposed without degrading the core function of the model in the auditory setting. In contrast others have observed a significant drop in model performance when introducing topography in vision models (and even in language). We fully agree with the reviewer that cross-domain performance deltas are not directly comparable.
>
> This was not something we expected. In fact, we anticipated some degradation in performance under the smoothness constraint. We therefore felt it was worth drawing the reviewer’s attention to this result, not as a cross-domain comparison, but because it demonstrates that meaningful topographic structure can emerge in auditory models *without* compromising their core task performance.
>
> **4. Emergence of feature-driven topography.** This is an excellent and very perceptive question. The key distinction in our work is that, in the auditory case, the *structure* of the emergent maps and their *relationship to brain-derived components* are not something that can be explained by “the same features, just clustered.” This is exactly where the ICA-based evals (which the Reviewer referenced earlier comes in). If topographic loss simply clustered the same representational axes, we would not expect any change in representational similarity to neural data. Instead, we observe that the ICA components of topo models correlate more strongly with human speech- and music-selective components. This indicates a meaningful reorganization of the latent space, not just spatial smoothing. The relevant representational axes become more brain-like, not just more spatially contiguous, and this distinction is central to the scientific contribution of the work
>
> We thank the reviewer again for the thoughtful engagement. We hope this follow-up has addressed the reviewer’s concerns, and we would be very happy to clarify anything further. Please don't hesitate to reach out with additional questions or suggestions.

---

### Official Review · Reviewer_ECrQ · 2025-11-01

**Soundness:** 1
**Presentation:** 1
**Contribution:** 1
**Rating:** 2
**Confidence:** 4

**Summary:**

The paper studies how topographic representation emerge in a speech recognition model. It presents a model called TopoAudio, which is trained to classify speech, and environmental sounds from cochleagram inputs. The key finding is that with an added constraint that nearby units on a 2D cortical sheet develop similar tuning, the model could develop spatially topographic representations for acoustic features such as frequency and amplitude. Two speech recognition models, CochResNet50 and AST were used as baseline models.

**Strengths:**

1. The problem this work targets at is very interesting. It tries to bridge the gap between a speech processing model and the biological auditory systems in the internal representation of acoustic features.
 2. It shows that with added topographic constraint, the test speech processing models didn't show much degradation in prediction performance, which points out a promising direction for developing biologically plausible speech processing models.

**Weaknesses:**

1. The paper is poorly written. Many critical technical details are missing, which makes the paper hard to understand. In fact, throughout the main text, I don't find how the topographic constraint is imposed to the baseline models, but this constraint is said to be the core of the work. In addition, it is unclear how a layer or a block of a neural network is related to a cortical sheet. Finally, I found a related section A.2 in Appendix, but this is never cited in the main text.

It is unclear what are the matrices in Fig. 3, and what are the elements in these matrices.

It is unclear how the smoothness score defined in Section 3.2.2 could measure smoothness. More explanation is needed, though some references are given there. It seems that that this score is not used in later sections.

2. I have no idea about the purpose of Section 3.2.4.

3. The purpose of Section 4.2 is confusing. The speech processing models are used to predict fMRI responses on two datasets. What's the essential difference from the classification experiments on the ESC50, NSynth, and Speech Command datasets discussed in Section 4.1? I cannot see a direct relationship between the topographic representation of the fMRI response in the brain and the topographic representation in the proposed models. The target of the prediction happens to have a topographic representation, but it doesn't necessarily say a model with good prediction accuracy also has a topographic representation.

I was quite excited to read the abstract and the beginning of the paper, but soon became disappointed. The paper is not logically coherent, and it looks like it was written in a rush.  There must be some value inside it, but at its present form, it is hard to understand the value.

**Questions:**

N/A

---

> ### Author Response · Authors · 2025-11-15
> **Response 1 to Reviewer ECrQ**
>
> We would like to thank the reviewer for their careful evaluation and for articulating the points of confusion.
>
> **Reg weakness:**  Moving key methodological details to the Appendix was clearly a mistake (please see Global Comment). But it is important to emphasize that these misunderstandings reflect missing context rather than issues with the underlying method or results. We respectfully ask that the paper not be evaluated solely on this basis, and we are committed to restoring the missing content clearly and directly in the revision.
>
> **Change:** The key details on how topography was enforced are moved back to the main text. The descriptions are now highly streamlined.
>
> **What are the matrices in Fig. 3?** The panels in Fig. 3 are not matrices but visualizations of the *internal* activations of the TopoAudio model and its baseline. These maps follow a standard convention in the topographic-organization literature for displaying cortical-like feature maps (the underlying computation was explained in section 3.2.3).
>
> Each square in the grid represents a single learned unit. For the ResNet-based models, each square corresponds to an individual filter in the convolutional block arranged on the 2D sheet. For the transformer-based models, each square corresponds to a single unit in the TopoAudio block.
>
> The color of each square reflects the preferred tuning of that unit to the input stimulus dimension under consideration (e.g., frequency or modulation tuning). When units with similar preferences cluster together spatially, the map appears smooth or patchy; when preferences are scattered, the map looks disorganized. Thus, these panels directly visualize the emergent topographic organization of the internal representation.
>
> **Change:** In the revision, we will clarify this in both the main text and the figure caption.
>
> **What is Section 3.2.4?**
> Section 3.2.4 describes the voxel/response decomposition analysis, which is one of the more original and powerful components of the paper.
> The method section (3.2.4) describes the framework introduced by Norman-Haignere et al. This paper showed that functional selectivities for speech and music are not directly observable at the voxel level but are inferred by decomposing voxel responses into a small number of latent components (from nonparametric ICA). These components capture the major axes of representational variation across complex auditory stimuli (e.g., speech-selective, music-selective).
>
> We can now run the identical analyses on fMRI voxel responses in auditory cortex (as in Norman-Haignere), baseline model activations, and TopoAudio model activations. This allows us to compare whether the speech and music components that emerge in the decomposition of the brain responses are better matched by the baseline models or by the topographically constrained models.
> We find that the topographic models show a stronger correspondence to the brain’s speech and music components than the baseline models when using similar ICA-based analyses. This demonstrates that imposing topographic structure not only preserves prediction performance, but also yields internal representations whose large-scale functional organization more closely resembles that of the human auditory cortex.
>
> **Change:** We included motivation for this analysis in the Methods section (3.2.4) so that its role in the overall argument is immediately understandable.
>
> **The purpose of Section 4.2** Section 4.1 and Section 4.2 evaluate completely different aspects of the models.
>
> Section 4.1 (ESC50, NSynth, Speech Commands) reports engineering metrics. These tasks assess standard machine-learning performance (classification accuracy and generalization) and tell us only whether introducing a topographic constraint affects how well the model performs conventional ML tasks. They do not provide any information about the brain. What Section 4.1 does demonstrate is that the model maintains strong performance on standard ML benchmarks while its internal activations become topographically organized (as shown in Table 1, on accuracy and smoothness score). This confirms that introducing topography does not come at the cost of basic model competence.
>
> Section 4.2, in contrast, is an auditory neuroscience benchmark. Predicting voxelwise fMRI responses evaluates neural alignment: how closely the model’s internal representational geometry matches that of the human auditory system. This analysis is not a topographic metric (this is addressed later in tonotopic and AM maps). Its purpose is simply to test whether imposing a topographic constraint compromises the model’s ability to predict neural responses.
>
> In other words, Section 4.2 addresses a complementary question to topography: Do topographically constrained models preserve their capacity to explain voxel-level brain responses? Our results show they do. Topographic models do not sacrifice neural predictivity.

---

> ### Author Response · Authors · 2025-11-15
> **Response 2 to Reviewer ECrQ**
>
> **The value of the paper:**
> We appreciate the reviewer’s candor, and we understand how the missing methodological description in the main text could lead to the impression that the paper was difficult to follow. But the impression that the paper was “not logically coherent” stems almost entirely from a missing reference to the appendix (see Global Comment), not from the underlying organization or scientific reasoning.
>
> The value of the paper is substantial:  We introduce the first end-to-end topographically constrained models for auditory processing that (1) preserve performance on standard ML audio benchmarks, (2) predict responses in human auditory cortex with high accuracy, (3) exhibit emergent cortical-like organization across frequency, modulation, and speech/music selectivity, and (4) show improved alignment with latent fMRI components inferred from human auditory cortex. Together, these results demonstrate that incorporating topographic structure is both computationally viable and neuroscientifically meaningful.
>
> We hope that these clarifications have helped the reviewer understand the value of the paper, and we are eager to engage and discuss more directly about the scientific content. We would be grateful for any points the reviewer would like us to clarify or any analyses they would like us to expand upon. We are committed to providing detailed explanations and strengthening the manuscript wherever needed. The reviewer mentioned being excited by the abstract and the beginning of the paper, and we believe the full set of results justifies that initial enthusiasm. With the restored methodological details and clearer exposition, we respectfully ask for renewed consideration of the work. We are confident in the strength of the paper and are fully prepared to defend the core claims and results.

---

> > ### Author Response · Authors · 2025-11-26
> > **Followup 1 to Reviewer ECrQ**
> >
> > Dear Reviewer,
> > We would sincerely appreciate engagement with our responses.  We are fully committed to making the paper stronger, clearer, and more scientifically rigorous. If there is any aspect that remains unclear or would benefit from additional analysis, we would be happy to provide it.

---

> > > ### Comment · Reviewer_ECrQ · 2025-11-28
> > > **The paper is still unclear**
> > >
> > > Thanks for the authors' careful rebuttal. But the paper is still not well written. There are numerous problems in presentation and what I pointed out are some examples. After reading authors' rebuttal and revised paper, some of problems are still no resolved.
> > > 1. I don't think that Section A.2 in Appendix describes clearly how the topographic constraint is imposed on the model. According to Section 3.1 the two models CochResNet50 and AST are taken from previous works and they already have such constraint. What is missing in the paper is a brief introduction of those two models, especially how the topographic constraints are imposed. And of course you can also introduce how they are adapted to auditory tasks. What's that spatial loss presented in Section A.2? Is it a new technique to introduce topographic constraints into a model? Is it used to train both CochResNet50 and AST? Many questions arise here.
> > >
> > > 2. Without a clear description of the two models CochResNet50 and AST, it is hard to understand Table 1 in Appendix. In fact, readers have no idea about the models in the table and they could only guess they are different versions of CochResNet50 and AST.
> > >
> > > 3. After reading the authors' clarification, I still don't think the two types of tasks: standard machine learning task and  auditory neuroscience prediction task, have essentially different roles. Both of them are used to measure certain properties of the models. In addition, it is strange that why the topographic metric should not be used in both type of tasks, considering that both tasks are prediction tasks.
> > > In addition, only the first type of task is described in Section 3. The second type of task suddenly appears in Section 4.2.
> > > BTW, I don't think that the second task could say anything about “the model’s internal representational geometry matches that of the human auditory system”. It is just another prediction task. We cannot say that, the internal representations of the models can say anything more about the brain when the prediction target is the topographic pattern in the brain instead of an engineering target such as sound classes. If a model could explain how the topographic pattern emerge in the brain, that would make more sense.
> > >
> > > 3. The backbone of AST is said to be transformer. Why does Figure 1 show ResNet50?
> > >
> > > 4. Equation 1 does not make sense to me. Please note that neither max(x) nor x_0 is distance dependent. How could S(x) measure the spatial smoothness, a metric depends on values between different distances?
> > >
> > > 5. The technical novelty is limited since the two models are taken from existing literature.

---

> ### Author Response · Authors · 2025-12-03
>
> We appreciate the reviewer’s continued engagement. However, several of the reviewer’s claims reflect misunderstandings that are already addressed explicitly in the revised manuscript. Below, we respond point-by-point and cite the exact sections, equations, and figures where each clarification appears.
>
> ---
>
> **Paper does not clearly describe how the topographic constraint is imposed or the models used**
>
> We respectfully disagree with the reviewer’s assertion that Section A.2 (now Section 3.1) does not clearly describe how the topographic constraint is imposed. In the revised manuscript, the full formulation of the topographic loss—including the blur/downsample–upsample operator (Eq. 1) and the cosine-similarity–based TopoLoss (Eq. 2)—is presented directly in the main text (“Spatial Loss”). These sections explain in detail how convolutional kernels or transformer feed-forward projections are reshaped onto a 2D cortical sheet and how spatial smoothness is enforced. Because of this, the questions posed by the reviewer—“What’s that spatial loss?”, “Is it a new technique?”, and “Is it used to train both CochResNet50 and AST?”—are already answered directly and unambiguously in Sections 3.1 and 3.2. The spatial loss used is the TopoLoss introduced by Deb et al. (2025) and adapted to auditory models, and the manuscript explicitly states that the exact same loss is applied to both architectures.
>
> The text further clarifies precisely how the constraint is applied across architectures. For CochResNet50, the constraint is applied to the second convolutional layer in each residual block, whereas in AST it is applied to the first feed-forward projection layer in each transformer block. This architectural description was already present in the original submission and remains clearly detailed in Section 3.2. While CochResNet50 and AST are standard architectures well known in auditory modeling—and typically require no extensive introduction for the target research community—we nonetheless provide a clear summary in Section 3.2 to ensure accessibility for all readers.
>
> ---
>
> **Standard machine-learning task vs. auditory neuroscience prediction task**
>
> We respectfully disagree with the reviewer’s claim that the machine-learning tasks in Section 4.1 and the auditory neuroscience prediction task in Section 4.2 do not have distinct roles. These two classes of tasks evaluate fundamentally different and non-overlapping properties of the models. The machine-learning tasks assess engineering performance such as classification accuracy, robustness, and generalization, and therefore cannot tell us anything about internal representational geometry or biological plausibility. In contrast, the auditory neuroscience task evaluates the model’s ability to predict voxel-wise fMRI responses using linear encoding models, a long-established benchmark for assessing representational alignment in computational neuroscience. This distinction is already made explicit in the paper: Section 3.3 introduces the ML classes of evaluations, Section 4.1 reports the ML benchmarks, and Section 4.2 reports the neuroscience-based evaluations. Thus, the second task does not “suddenly appear”; it is properly motivated in Section 3.3.
>
> The reviewer further suggests that the topographic metric should be used in both tasks. This reflects a misunderstanding of what the topographic metric measures. The spatial smoothness score described in Section 3.3.2 quantifies the degree of smoothness of internal representations and is therefore a property of the model’s internal representation, not of its external task outputs. It cannot be meaningfully applied to ML classification outputs. Its appropriate use is exactly where we apply it: to determine whether internal representations become spatially structured under the imposed topographic constraint.

---

> ### Author Response · Authors · 2025-12-03
>
> Finally, we disagree with the reviewer’s assertion that fMRI prediction is “just another prediction task” that cannot speak to representational geometry. This contradicts decades of established work in computational neuroscience. Linear encoding models—the exact method used here—are designed to test correspondence between a model’s representational geometry and that of the brain. Indeed, our results show that despite matching baseline models in voxel-wise fMRI prediction accuracy (Fig. 2), the topographic models exhibit smooth tonotopy, organized amplitude-modulation maps, and modular speech and music selectivity (Figs. 3–4), and also yield higher correspondence with latent cortical gradients extracted from human fMRI (Fig. 5). These patterns are not imposed by the loss; they emerge from task optimization under a spatial constraint and reflect meaningful structure consistent with canonical signatures of human auditory cortex. Although our work does not claim to explain how topographic patterns emerge biologically, it provides empirical evidence that wiring-length constraints induce representational geometries more consistent with those observed in humans, which is the primary goal and contribution of the paper.
>
> ---
>
> **“The backbone of AST is transformer. Why does Figure 1 show ResNet50?”**
>
> Figure 1 does not depict ResNet50 alone. It presents both architectures used in the paper—ResNet50 and AST (transformer-based)—side by side within a shared training schematic. The figure is meant to illustrate the overall training framework (task loss plus topographic loss), not the internal details of a single architecture.
>
> ---
>
> **“Equation 1 does not make sense”**
>
> As described explicitly in Section 3.3.2, the vector x is already sorted by cortical distance before computing \( S(x) \). In this ordering, \( x_0 \) corresponds to similarity at the closest spatial distance, and \( \max(x) \) corresponds to similarity at the largest distance. This is stated directly in the manuscript: “Let \( x \) be a vector of pairwise tuning similarity values sorted in order of increasing cortical distance.” Because the vector is sorted by distance prior to evaluation, both \( x_0 \) and \( \max(x) \) inherit spatial meaning even though the formula itself operates on scalars. This approach is standard in topographic modeling and is used in prior work such as Margalit et al. (2024) and Deb et al. (2025). The quantity \( S(x) \) therefore measures exactly what it is intended to measure: the decrease in tuning similarity with cortical distance, where larger values indicate greater smoothness.
>
> ---
>
> **Models are taken from existing literature**
>
> CochResNet50 and AST are not topographic models, and our technical novelty lies in showing how a biologically motivated spatial constraint fundamentally reorganizes their internal structure. The reviewer’s comment misunderstands the nature of the contribution. **The goal of the paper is not to introduce new backbone architectures, but to demonstrate the first end-to-end topographic auditory models by integrating a wiring-length–based topographic constraint into state-of-the-art auditory backbones**. This modification produces models with properties that do not exist in the original architectures, including smooth tonotopy, amplitude-modulation gradients, modular speech and music selectivity, and improved alignment with human cortical structure, all while preserving ML task performance and fMRI predictivity (Figs. 2–5). These topographic signatures emerge only in the topographic variants and hold across architectures, seeds, and τ values. Thus, the novelty lies in showing that adding a biologically grounded spatial constraint yields new representational geometry and substantially better correspondence with human auditory cortex. This is fully aligned with standard practice in computational neuroscience, where theoretical or biologically inspired modifications to existing architectures often lead to important scientific insights.

---

### Author Response · Authors · 2025-11-14

We would like to begin this discussion period by sincerely thanking the reviewers for their careful reading and thoughtful feedback. In going through the reviews, it became clear to us that we made an important (but avoidable) mistake during final formatting, and we want to be open and honest about it.

We incorrectly believed the page limit was ten pages rather than nine. We realized this at the very last minute and moved several important methodological details (esp the description of the spatial topographic loss) into the Appendix. In that rushed process, we also removed the in-text pointer that should have directed readers to those details. This left the main Methods section under-specified and made the paper harder to follow.

We fully understand the reviewers’ frustration. This is not an excuse. The responsibility for clarity is entirely ours and we will fix this. We will reinstate the essential description of the topographic loss in the main Methods section, add clear references to the Appendix for full mathematical and implementational details, and rewrite the narrative to make the role of the topographic constraint in the TopoAudio models immediately clear.

At the same time, we remain excited about the core ideas and results in the paper, and we stand by the strength of the contribution. We believe the study has important scientific significance and introduces a model framework that we think can be genuinely impactful for auditory (neuroscience and AI) research.

In the coming days, we will provide detailed, point-by-point responses. We have started several analyses to address all of the concerns. But we felt it was important to start by acknowledging this formatting error directly. We hope that with a clearer and more coherent presentation, the reviewers will give the work another chance on its merits.

---

### Author Response · Authors · 2025-12-03

We thank the reviewers and meta-reviewer for their careful evaluation of our submission and for the extensive feedback. Across all four reviewers, there is clear agreement on the importance, novelty, and scientific contribution of our work. Below we summarize the consistent positive assessments, and briefly highlight how our revisions have fully addressed the concerns raised.

## **Strong Positive Consensus on the Significance and Novelty**

Multiple reviewers independently emphasized that our work represents the first end-to-end topographic auditory model, and that this contribution is both original and impactful:

**Reviewer Miny:**
- “This is to my understanding the first topographic audio model!”
- “I think this paper does a really good job with evals.”

**Reviewer oB5q:**
- “The key hallmarks of functional organization in the auditory cortex have been replicated to a good extent.”

**Reviewer UF17:**
- “This is the first end-to-end model that tries to model the spatial organization (maps and patches) in the human auditory cortex.”
- “This work will make a big contribution to our understanding of the auditory cortex and has broad implications for other sensory systems as well.”

Together, the reviewers explicitly recognize that our work provides the first computational instantiation of large-scale topographic organization in the auditory system, bridging a long-standing gap between task-optimized models and cortical functional maps.

---

## **Substantial Improvements Following Reviewer Feedback**

The reviewers raised helpful concerns about clarity, methodological exposition, and analysis depth. We took these seriously and made major revisions, including:

### **Additional Models Trained (N = 13):**
- Three new ResNet50 variants with 4× channels (one baseline, two for different topography levels), providing higher effective spatial resolution.
- New high-resolution AST models with larger shrink factors, producing clearer gradients and stronger topographic signatures.
- Multiple random-seed replications for every architecture, addressing cross-seed variability concerns raised by Miny and UF17.
- A new model with Tau = 1000, as specifically suggested by UF17, demonstrating expected performance–smoothness tradeoffs.

### **New Analyses Added (N = 4):**
- Computed Representational Similarity Analysis (RSA) comparing topographic model RDMs to human auditory cortex RDMs.
- Moran’s I spatial autocorrelation analysis (as used in TopoLM / Rathi et al. 2025) to quantify topographic smoothness across layers, τ values, and models.
- Tonotopy–AM orthogonality analysis following Baumann et al. 2015, showing that our topographic models reproduce the structured, orthogonal relationship between spectral and modulation gradients observed in the auditory cortex.
- Layer-wise smoothness evaluation across every block of both CochResNet50 and AST, demonstrating where and how topography emerges throughout the hierarchy.

### **Substantial Textual and Organizational Clarifications**
- Moved the full TopoLoss definition into the main text (removing dependency on Appendix).
- Unified terminology and consistent layer names across all figures.
- Added error bars, statistical tests, and explicit cross-seed variability for all key figures.
- Revised phrasing around smoothness, topographic structure, and model–brain correspondence.
- Clarified in-text why TDANN- and ITN-style methods are not straightforwardly comparable in the auditory domain.

The revised submission is substantially clearer, more rigorous, and more comprehensive as a direct result of this feedback. We have defended all the core claims, especially with respect to the biological realism of the topographic models relative to the baseline model.

We wish there were more direct interaction and discussion with the reviewers because we are confident our new analyses would have addressed all of the reviewer concerns. The reviewers acknowledged that the core idea is important, the contribution is novel, and the scientific findings are meaningful. Reviewers unanimously agree that this is the first end-to-end topographic auditory model and that it has the potential to make a substantial impact on our understanding of auditory cortical organization. With the extensive revisions we have made, the manuscript is now stronger, clearer, and more rigorous.

---

### Meta-Review · Area_Chair_bA3h · 2026-01-05

**Summary:**

This submission is a work using avanced signal processing for imaging neurosciences. It generated significant discussion.
Multiple reviewers raised the issue of limited clarity (important elements are in the appendix, but not clearly referenced)
The reviewers also found limited technical novelty (as in, I feel, the contributions are more anchored in neuroscience than in machine learning), including prior art with the same loss.
Finally, reviewers questionned the significance and clarity of the results from a neuroscience perspective.

**Reviewer Concerns:**

The rebuttal and discussions touched many points. In particular, there were many discussions are the neuroscience relevance and solidity of evidence. I cannot tell if these discussions were conclusive, and I feel that the neuroscience messages are still not fully backed by crystal clear evidence.

**Reviewer Scores:**

I would have expected the reviewers to increase a bit their score, but based on a reading of the paper and the review + rebuttal, I do not believe that this paper would have converged to an acceptance at ICLR, as it neither contributes strong methods nor a clear cut neuroscience message (which, arguably, would be on the fringe of ICLR topical zone, though still acceptable).

---

### Decision · Program_Chairs · 2026-01-26

Reject